# Probing the Impact of Scale on Data-Efficient, Generalist Transformer World Models for Atari

**Jooyeon Kim**                                          *jooyeon.kim@unist.ac.kr*
*Graduate School of Artificial Intelligence*
*UNIST*

**Reviewed on OpenReview:** *https://openreview.net/forum?id=wVcvqtKaMY*

## Abstract

Developing generalist systems that retain human-like data efficiency is a central challenge. While world models (WMs) offer a promising path, existing research often conflates architectural mechanisms with the independent impact of model *scale*. In this work, we use a minimalist transformer world model to analyze scaling behaviors on the Atari 100k benchmark, using fixed offline datasets derived from a presupposed expert policy. Our results reveal that environments fundamentally fall into distinct scaling regimes, even when constrained by identical offline data budgets and model capacities. For individual tasks, some environments naturally allow models to pass the interpolation threshold, yielding monotonic improvements in the overparameterized regime, while others remain trapped in the classical regime, where larger world models degrade fidelity. In the unified setting, i.e., a single transformer trained on a suite of 26 Atari environments, we uncover that joint training stabilizes scaling dynamics, ensuring monotonic gains across all environments, regardless of their distinct inherent scaling regimes. Finally, we demonstrate that improved fidelity translates directly to downstream control, with policies learned entirely within the simulated dynamics achieving a median expert-random-normalized score of 0.770. Our findings suggest that future progress lies as much in precise scaling strategies as in architectural innovation.

## 1 Introduction

The development of generalist artificial intelligence (AI)—a single system capable of mastering a diverse suite of tasks with a *unified* set of parameters—remains a central goal of AI research (Reed et al., 2022). While large-scale foundation models have successfully realized this unified approach (Bommasani et al., 2021), their ability to scale monotonically—where larger capacity reliably yields better performance (Kaplan et al., 2020)—is reliant on data abundance (Hoffmann et al., 2022), optimization procedures (Loshchilov & Hutter, 2019; Yang et al., 2021; You et al., 2020), and regularization (Neyshabur et al., 2014; Soudry et al., 2018; Lin et al., 2024). However, such a data-hungry paradigm stands in stark contrast to human intelligence, which grasps complex dynamics from highly limited interactions (Lake et al., 2015). Consequently, an underexplored challenge is understanding how scaling behaviors fundamentally shift in data-scarce settings. Mapping these dynamics is essential to unlock the benefits of model scale without the prerequisite of massive datasets, ultimately paving the way for highly sample-efficient generalist AI.

The Atari 100k benchmark serves as a critical testbed for this pursuit. Constrained to a total budget of 400k environmental frames, corresponding to 100k agent interactions with a frame skip of 4, the benchmark limits experience to roughly two hours of human gameplay, thereby qualifying as a strictly data-efficient setting. Furthermore, the benchmark offers a suite of distinctive dynamical systems across 26 games, demanding a diverse array of capabilities: from reaction speed in Asterix and motor precision in Freeway, to path-planning and spatial awareness in MsPacman, risk assessment in Seaquest, and delayed gratification in Pong.

World models (WMs) (Ha & Schmidhuber, 2018a;b) offer a promising path toward such data-efficient, generalist AI. By learning a generative model of the environment, WMs enable planning within virtual rollouts, analogous to human imagination (Hamrick, 2019). This paradigm shifts the burden of sample complexity; environmental interaction is devoted solely to forming the model, while the extensive trial-and-error process of policy learning occurs entirely within the learned dynamics, incurring no real-world cost.

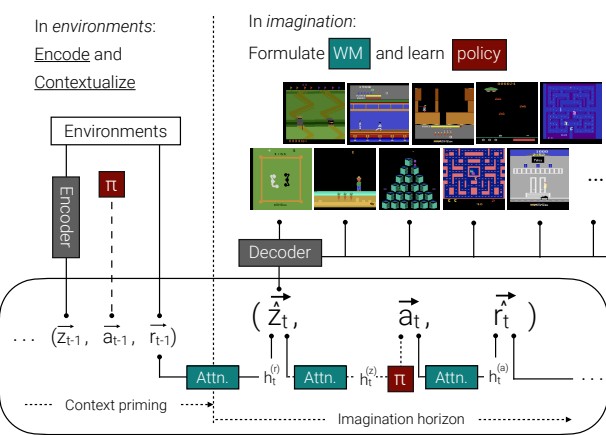

Figure 1: Overview of the minimalistic world model. **(Left)** We collect sequences of triplets (latent embeddings $\vec{z}_{t-1}$, actions $\vec{a}_{t-1}$, rewards $\vec{r}_{t-1}$) to construct a fixed offline dataset. No policy learning occurs during this interaction phase. **(Right)** Unrolling the imagination. Using the offline data as context, the model autoregressively generates future tokens $\tilde{\vec{z}}_t$ and $\tilde{\vec{r}}_t$, and the policy is learned inside the imagined, virtual roll-outs to sample the action $\vec{a}_t$.

Consequently, a substantial body of work has proposed achieving data-efficient or generalist agents using WMs in the Atari domain. Aside from recent probabilistic approaches (Alonso et al., 2024), the dominant backbone for these systems remains the transformer (Vaswani et al., 2017). Methods such as the transformer world model (TWM) (Robine et al., 2023), IRIS (Micheli et al., 2023), and STORM (Zhang et al., 2023) have demonstrated remarkable data efficiency, while Multi-Game Decision Transformers (Lee et al., 2022) have explored generalist capabilities across large suites of gaming dynamics.

Despite this progress, the current landscape lacks a rigorous analysis of *scale*. Existing literature prioritizes novel architectural mechanisms and structural priors, yet their implementations, e.g., model depth, width, and context length, vary drastically (Refer to Appendix F for our summary). This conflation of methodological innovation with arbitrary scaling choices obscures the precise drivers of empirical improvements. Consequently, it remains unclear whether the performance saturation observed in complex environments stems from inherent architectural deficiencies or simply from operating within sub-optimal scaling regimes.

In this work, we adopt a minimalist approach to devise a transformer world model using standard, off-the-shelf components (section 3), allowing us to conduct a controlled analysis on the impact of scale for both individual (environment-wise) and unified world models (all 26 environments). Our primary findings are:

- In subsection 4.1, we establish a robust performance benchmark by constructing a fixed offline dataset for each environment using a presupposed expert policy. This allows us to evaluate the fidelity of the world model—formulated strictly from this offline data—by measuring the extent to which a policy trained entirely within the learned dynamics can recover or surpass the performance of the presupposed policy.

- In subsection 4.2, we demonstrate that optimal scaling for individual world models is strictly task-dependent, with different environments naturally falling into distinct scaling regimes. Even within a fixed offline data budget, environments exhibit distinct scaling behaviors; certain environments allow models to pass the interpolation threshold, resulting in monotonic performance improvements, while others remain trapped in a classical parameterization regime, i.e., *bigger models hurt.*

- In subsection 4.3, we examine the unified world model trained on all 26 games. In direct contrast to diverging scaling patterns observed in individual models, the unified training stabilizes the overall scaling behavior, ensuring that increasing model capacity, after surpassing the interpolation threshold, yields monotonic improvements across all individual tasks regardless of their original scaling regimes, i.e., *bigger models never hurt.*

- In subsection 4.4, we demonstrate that the improved fidelity achieved solely through proper scaling—isolated from modeling idiosyncrasies—translates directly to effective downstream policy learning. We show that policies learned entirely inside our minimalist, properly scaled world models achieve the median normalized scores of 0.770 relative to the expert performance of one and the random performance of zero, with the exact same model configurations and environmental steps as the presupposed expert policy.

Overall, we highlight the critical role of scale in generative world models for achieving data-efficient generalist AI, suggesting that future progress may depend as much on scaling strategies as on architectural innovations.

## 2 Related work

World models (Ha & Schmidhuber, 2018a;b) learn to simulate environmental dynamics, enabling agents to plan actions based on synthesized future trajectories rather than through environmental interactions. Crucially, this paradigm extends to training agents entirely within the imagined Markov decision process (MDP) instantiated by itself—a process often referred to as learning in imagination (Hamrick, 2019).

To improve the training outcomes using the imagination MDPs, we need a stable world model that accurately mimics the real environment and generalizes well to yet unexplored parts of the world. The Dreamer models and their variants (Hafner et al., 2019; 2021; Ma et al., 2024; Hafner et al., 2025a) build on recurrent state-space models (RSSMs) to form world models and operate on continuous (DreamerV1) or discrete (DreamerV2, DreamerV3) latent representations. Notably, DreamerV3 uses fixed hyperparameter settings for different Atari games and other domains, but uses separate models (model parameters) and policy learning algorithms for different environments. Łukasz Kaiser et al. (2020) propose SimPLe that uses long short-term memory (LSTM) (Hochreiter & Schmidhuber, 1997) to model sequences of discrete latent representations.

Aside from the above models, Chen et al. (2022) propose transformers (Vaswani et al., 2017) as world model backbones. Micheli et al. (2023), Robine et al. (2023), Micheli et al. (2024), and Zhang et al. (2024b) propose transformer-based world models that outperform other model-based and model-free approaches on Atari 100k (Łukasz Kaiser et al., 2020) without lookahead search. Burchi & Timofte (2025) further advance this with TWISTER, incorporating contrastive predictive coding to enhance long-horizon consistency in transformer world models. Similarly, Agarwal et al. (2024) propose a transformer-based world model with discrete input tokens using a vector-quantized VAE (Van Den Oord et al., 2017) and use the decoder to retrieve inputs for the learning algorithm. A primary benefit of employing discrete embeddings via VQ-VAE, as seen in Micheli et al. (2023) and Agarwal et al. (2024), is the effective capture of visual details by distributing information over multiple tokens. Furthermore, discrete tokenization fits seamlessly into standard NLP transformers, mirroring the inherent discreteness of human language. The downside is slow inference in the world model, as generating multiple tokens per image results in an increased number of function evaluations (NFE). Alternatively, Hafner et al. (2021; 2025a); Robine et al. (2023); Zhang et al. (2024b) build upon the categorical encoder-decoder, which enhances generalization capability through latent sparsity (Hafner et al., 2021). When applied to transformer world models, it allows for assigning a single token per image observation (Zhang et al., 2024b), thereby reducing inference cost. Refer to Appendix B.1 for an extensive discussion on preserving the visual details in different world model architectures.

Departing from discrete tokenization, Hafner et al. (2025b) introduce DreamerV4, which utilizes a block-causal transformer operating on a grid of continuous latents. To overcome the inference bottleneck of autoregressive generation, they propose a diffusion-based objective termed shortcut forcing, a variant of the diffusion forcing (Chen et al., 2024), which enables the model to predict future states with flexible step sizes. This approach reconciles the trade-off between spatial fidelity and computational efficiency, allowing for real-time imagination of high-dimensional dynamics without the cost of sequential token prediction.

Lee et al. (2022) proposes the multi-game decision transformer that trains a single agent capable of solving multiple Atari games. They split a single image observation into patches and use uniform quantization for image tokenization. Contrary to our work, it requires hundreds of millions of data points or relies on human guidance to train a generalist agent. Nauman et al. (2024) demonstrate that increasing model capacity, combined with categorical value distributions and regularization, enables efficient multi-task learning on Atari. Similarly, Schwarzer et al. (2023) show that in the model-free setting (BBF), scaling the network depth and width is a primary driver of data efficiency, outperforming complex algorithmic modifications. Cheng et al. (2025) also highlight the importance of scale in the offline setting, proposing JOWA to jointly optimize world and action models for improved performance. Deng et al. (2023) adopt S4 models (Gu et al., 2022) that excels in modeling long-term dependencies and fast inference. Zhang et al. (2024a), Ding et al. (2024) and Alonso et al. (2024) concurrently propose diffusion (Ho et al., 2020)-based world models, which excel in capturing visual details at the cost of lacking the capability to model long-term dependencies.

In contrast to prior works that rely on novel inductive biases to enhance fidelity, we adopt a strictly minimalist design philosophy. We purposefully employ standard, off-the-shelf components to isolate the effects of model scale necessitated by diverse Atari gaming environments. This work thus serves as a validation of model scale rather than architectural idiosyncrasies in the regime of data-efficient generalist world models.

# 3 Method

We consider a multi-environment Markov decision process (MEMDP) (Raskin & Sankur, 2014), which generalizes a partially observable Markov decision process (POMDP) (Kaelbling et al., 1998; Sutton & Barto, 1998) to a setting with distinct underlying dynamics. Formally, an MEMDP is defined as a tuple $\langle \mathbb{S}, \mathbb{A}, \mathbb{O}, \{T_k\}_{k=1}^K, R, Z, \gamma \rangle$. Here, $\mathbb{S}$ denotes a set of states, $\mathbb{A}$ is a set of discrete actions, and $\mathbb{O}$ represents a set of RGB pixel-valued image observations. There are $K$ transition functions, $T_k : \mathbb{S} \times \mathbb{A} \times \mathbb{S} \to [0,1]$, each of which describing the environment dynamics $p_k(s_{t+1}|s_t, a_t)$, as well as the reward function $R : \mathbb{S} \times \mathbb{A} \to \mathbb{R}$. Finally, $Z$ creates links from the states to the observations, i.e., $Z : \mathbb{S} \times \mathbb{O} \to [0,1]$. We aim to optimize a policy $\pi$, taking states as inputs and outputs actions, to maximize the expected return $\mathbb{E}[\Sigma_{t \geq 0} \gamma^t r_t]$, with $r_t$ being the reward at time $t$ and $\gamma \in [0,1]$ being the discount factor.

## 3.1 Background

World models (Ha & Schmidhuber, 2018a;b) are "generative models of environments" (Alonso et al., 2024). The world model's objective is to learn a probabilistic mapping of the environmental dynamics, $p(s_{t+1}, r_t|s_t, a_t)$, using the past traces of an agent's interactions with the environment. As the world model closely approximates the real environment, it can accurately infer both future observations and reward signals that the environment is likely to generate from a specific sequence of actions. By repeating this inference procedure over multiple time steps in an autoregressive manner, one can derive virtual rollouts of the future, commonly and conveniently referred to as imagination (Hamrick, 2019).

From the perspective of policy learning, the practical merit of using well-trained world models is sample efficiency. As long as the world model closely reproduces the environment, an agent becomes capable of optimizing a policy solely within the imagination unrolled by the world model without needing to adjust its policy directly through repetitively making trials and errors in the real environment. Interacting with the environment is necessary only to formulate and solidify the world model.

The standard formulation (Ha & Schmidhuber, 2018a) of the world model comprises three distinctive modules: 1) a visual sensory component $V$ that encodes high-dimensional observations into stochastic latent embeddings, 2) a memory component $M$ that predicts the future given the past traces of observation-action-reward triplets, and 3) a decision-making component $C$ that optimizes the policy inside the imagination MDP, i.e., the virtual rollouts generated by $M$ and $V$ up to a finite imagination horizon.

## 3.2 World model formulation

We present a world model formulation designed to capture the diverse dynamics of Atari environments with sufficient fidelity to enable seamless zero-shot policy transfer from the world model to the real environments. Adhering to the standard paradigm (Ha & Schmidhuber, 2018b), our framework decomposes the agent into the three aforementioned distinct components. To this end, we follow the standard world-modeling scheme.

**Visual sensory component ($V$).** Following the formulation in Ha & Schmidhuber (2018a), we employ a vanilla variational autoencoder (VAE) (Kingma & Welling, 2014; Rezende et al., 2014) that compresses high-dimensional observations into a compact, continuous latent space modeled by multivariate Gaussian distributions. VAE encodes observations into the latent embeddings, i.e., $z_t \sim q_\phi(z_t|o_t)$, with $q_\phi(z_t|o_t)$ being the recognition model, i.e., the variational distribution that approximates the underlying posterior and $\phi$ being the neural network parameterization. To permit gradient-based optimization through this stochastic node, VAE adopts the reparameterization trick, expressing $z_t$ as a deterministic transformation of the encoder outputs and independent noise. Subsequently, the probabilistic decoder corresponds to $o_t \sim p_\theta(o_t|z_t)$, with $\theta$ being the respective set of neural network parameters of the decoder. Here, a single VAE encoder encodes the observations generated by multiple environments. Refer to Appendix B for comprehensive details on model architecture, training configurations. We provide comparisons with alternative visual backbones used in comparable world model approaches that are operationalized on the Atari 100k benchmark in Appendix B.1.

**Memory component ($M$).** We adopt a GPT-style (Radford et al., 2019) decoder-only transformer (Vaswani et al., 2017) as the architectural backbone of the memory component, with the latent embedding dimensionality of $d$ and $L$ layers. The model processes a sequential trajectory of triplets $(z_t, a_t, r_t)$, where $z_t$ is the continuous latent embedding sampled from the VAE, $a_t$ is the discrete action token, and $r_t$ is the reward. Following prior approaches (Micheli et al., 2023; Alonso et al., 2024), we discretize the reward signal by applying the sign function to $\{-1, 0, 1\}$ and combining it with the termination signal, resulting in a vocabulary of six distinct categories. This formulation necessitates a hybrid modeling approach, handling continuous latent embeddings $z_t$ alongside discrete tokens representing both actions and rewards.

To predict the next latent state $z_t$, we first compute the contextualized transformer representation at the reward token of the previous time step $r_{t-1}$:

$$h(r_{t-1}) = \text{TRANSFORMER}(z_{<t}, a_{<t}, r_{<t}).$$

We then pass this context vector through a multi-layer perceptron (MLP) projection head to compute the estimate $\hat{z}_t = \text{MLP}(h(r_{t-1}))$, minimizing the Mean Squared Error (MSE) between $z_t$ and $\hat{z}_t$. Subsequently, to predict the current reward $r_t$, we compute the context vector at the current action $a_t$, which incorporates the information from the current latent state:

$$h(a_t) = \text{TRANSFORMER}(z_{\leq t}, a_{\leq t}, r_{<t}).$$

This is passed to an MLP to compute the estimate $\hat{r}_t = \text{MLP}(h(a_t))$, which is optimized using cross-entropy (CE) loss. The final training objective is a weighted sum of these components, computed as $\mathcal{L} = \mathcal{L}_{\text{MSE}} + \alpha \cdot \mathcal{L}_{\text{CE}}$, where $\alpha$ is a configurable scalar hyperparameter regulating the trade-off. Refer to Appendix C for additional details on the transformer world model backbone.

**Decision-making component ($C$).** Policy optimization occurs entirely within the imagined MDPs. While the memory component predicts the environmental dynamics $(\hat{z}_t, \hat{r}_t)$, the policy $\pi$ determines the action $a_t$. We project the predicted latent state $\hat{z}_t$ to an observation embedding $\hat{o}_t$ via an MLP from the transformer and apply frame stacking to capture temporal context:

$$\hat{o}_t = \text{MLP}(\hat{z}_t), \quad a_t \sim \pi(a_t \mid \hat{o}_{\leq t}).$$

Distinct from prior works utilizing Soft Actor-Critic or Dreamer-style objectives, we employ Proximal Policy Optimization (PPO) (Schulman et al., 2017). This choice is strategic: since the expert data was generated via PPO, retaining the identical algorithmic structure isolates world model fidelity as the sole variable of interest. This allows us to rigorously assess the extent to which the learned dynamics are accurate enough to recover the expert's performance benchmark, which is learned exclusively inside the real environments, with hundreds of millions of interactions, without algorithmic, structural, and configurative confounds. Refer to Appendix A for implementation details on the presupposed, expert PPO policy.

**Remark.** We emphasize that the adoption of this *minimalist design philosophy*—eschewing complex auxiliary objectives or domain-specific inductive biases—is a deliberate design choice. Our primary objective is to investigate the feasibility of a *unified* world model capable of generating imaginary MDPs with sufficient fidelity across multiple Atari environments, constrained by a limited data budget of 100k samples per task. By stripping the architecture down to its essentials and employing standard off-the-shelf components, we establish a clean, controlled setting to isolate the fundamental effects of model scale and task variation on capturing both environment-specific dynamics and general intricacies that permeate all games. This ensures that the observed capabilities of the unified agent are attributable to the underlying capacity of the model rather than architectural idiosyncrasies. Throughout our experiments and in Appendix B, we empirically validate that this minimalist setup suffices to resolve essential visual details and dynamics, enabling the policy learned solely within the world model to asymptotically recover the performance benchmark of the offline dataset, on which the world model itself was formulated, when deployed in the real environment.

## 4 Experiments

### 4.1 Experimental setup

**Construction of the offline data testbed.** We construct a fixed offline dataset to formulate and evaluate both environment-wise individual and unified world models. Adhering to the data budget constraints of the Atari 100k benchmark (Łukasz Kaiser et al., 2020), we limit the interaction history for each of the 26 environments to 400k environmental steps. With a standard frame-skip parameter of 4, this amounts to 100k transitions, stored as quadruplets of observation, action, reward, and termination signals. Unlike the standard online setting, where agents continuously update their policies against the real environment, our protocol restricts the world model to this static dataset. This design creates a controlled testbed to rigorously assess whether model scaling alone can capture the diverse dynamics required for generalist performance.

**Dataset composition and the presupposed policy.** The composition of this offline dataset is critical. A dataset collected via purely random policies fails to traverse the later stages of complex, multi-stage games such as Hero, Frostbite, or Qbert, effectively depriving the world model of the experience required to model downstream environmental progression. Conversely, a dataset collected exclusively by a deterministic expert policy lacks the necessary stochasticity; a world model trained on such narrow trajectories typically collapses when the imaginary agent deviates even slightly from the expert's path. To balance the need for deep state-space coverage with robust error recovery, we employ a policy-based data collection strategy using the Proximal Policy Optimization (PPO) algorithm (Schulman et al., 2017). We train these PPO agents from scratch for all 26 environments rather than using publicly available checkpoints. This decision is twofold: first, it ensures complete coverage of all 26 games; second, and more importantly, it allows us to enforce an identical architectural configuration for both the data-collecting PPO and the policy optimization algorithm used later inside the world model. By aligning the model structures, we eliminate potential confounding variables arising from algorithmic discrepancies, allowing us to verify the degree to which the world model can recover the asymptotic performance benchmark of the original data collection policy.

**Collection schedule and stochasticity injection.** To capture a comprehensive range of environmental dynamics, we implement a collection schedule with gradually decreasing stochasticity. Using the pre-trained PPO checkpoints, we collect trajectories where the agent selects a random action with probability $p_{\text{rand}}$ and follows the expert policy otherwise. The noise parameter decays according to the schedule $p_{\text{rand}} = 1 - \frac{\log_{10}(1+i)}{5}$ at collection step $i$, transitioning from high-entropy exploration in the early phase to near-deterministic expert behavior as $i$ approaches 100k. Configuration details on the PPO training hyper-parameters, specific operation conditions, and environmental preprocessing are provided in Appendix A.

**Configurations.** Detailed model and training configurations, learning curves, and the encode-decode results on the 26 Atari gaming environments are covered in Appendix B. The model and training configurations, as well as the environment-wise learning curves, are disclosed in Appendix C.

### 4.2 Results on individual world models

We analyze the performance of individually trained world models, denoted as $\mathcal{W}_{\text{ind}}^{(\text{Env})}$, where the superscript specifies the target environment (e.g., $\mathcal{W}_{\text{ind}}^{(\text{Alien})}$). Our objective is to characterize how diverse environmental dynamics interact with fixed model capacities. Specifically, we investigate whether different gaming environments exhibit distinct scaling behaviors—effectively occupying different parameterization regimes—despite sharing identical data budgets and architectural configurations. We hypothesize that these variations arise because individual environments naturally fall into distinct scaling regimes. To empirically verify this, we train transformer world models for each environment with a fixed embedding dimension of 512 while varying the network depth $L \in \{2, 4, 8, 12, 24, 48, 96\}$. This sweep results in a linear progression of model capacity, ranging from approximately 6 million to 300 million parameters.

Figure 2 visualizes this landscape. The top panels illustrate the schematic of the deep double descent phenomenon (Nakkiran et al., 2021), mapping generalization risk against relative model complexity. In the

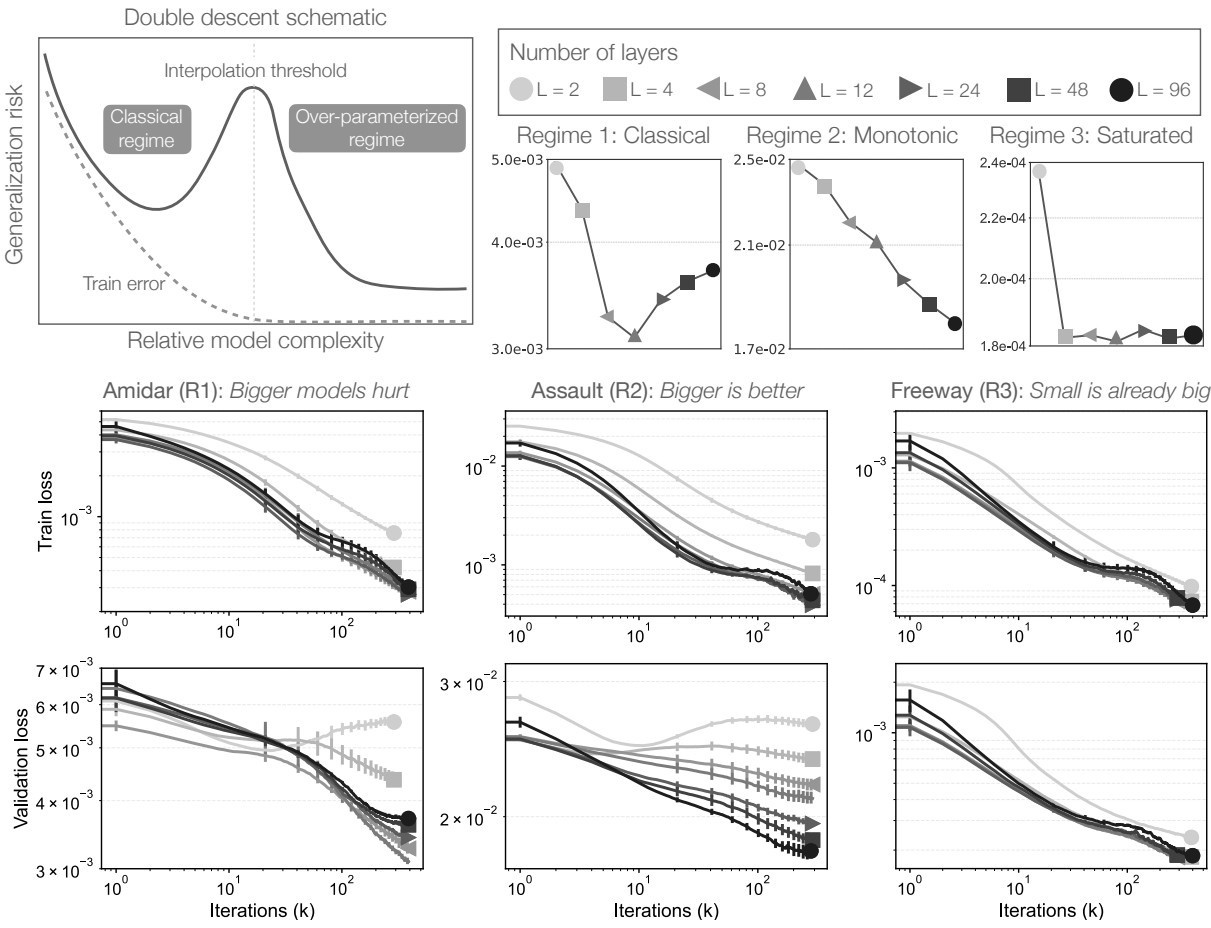

Figure 2: **(Top left)** Schematic of the deep double descent phenomenon with the generalization risk peaking at the interpolation threshold, which dichotomizes the classical and modern overparameterized regimes. **(Top right)** Divergent generalization regimes. Even with the identical sample budget ($N = 10^5$) and model configurations ($L = 2 \dots 96$), increasing model size yields different trends depending on task variations. Certain tasks remain in the classical regime (*Bigger models hurt*), while other tasks benefit from over-parameterization (*Bigger is better*), or saturate immediately (*Small is already big*). **(Bottom)**: Empirical loss curves. Model depths are distinguished by colors and markers, from $L = 2$ (green circle), $L = 4$ (light gray square), up to $L = 48$ (black square), $L = 96$ (red circle). Validation trends of three different Atari environments confirm the three aforementioned regimes: for certain tasks, such as Amidar, larger models generalize worse, with $L = 8$ outperforming $L = 96$; for another group of environments, such as Assault, performance improves monotonically with model size; and the other group,s such as Freeway, curves overlap, indicating saturation at $L = 4$. We present results averaged across eight independent runs and use a $\alpha$-trimmed mean filter with $\alpha = 0.2$ and a window size of 10. Error bars represent one standard error.

classical regime, increasing model capacity initially reduces bias but eventually leads to variance-dominated overfitting, characterized by a peak in the generalization risk at the interpolation threshold. Beyond this lies the over-parameterized regime, where increasing model size further acts as a robust regularizer, driving generalization risk down once again. Crucially, for a fixed-size offline dataset, the exact position of this interpolation threshold is highly task-dependent. Identical model architectures applied to different environments can exhibit distinct scaling regimes; a model size that surpasses the interpolation threshold in one environment may remain trapped in the variance-dominated classical regime when applied to another.

The bottom panels of Figure 2 confirm this by analyzing representative environments that exemplify the classical, monotonic, and saturated scaling regimes. We observe distinct scaling behaviors for each category:

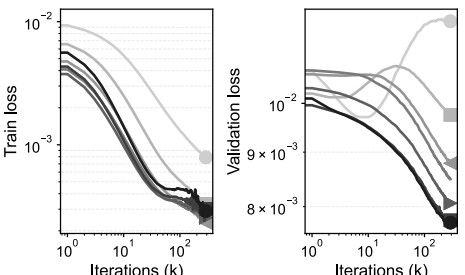 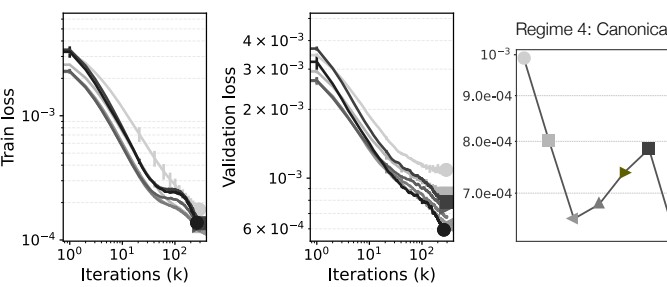

Figure 3: **(Left)** Average loss curves across 26 separately-trained transformer world models, each with its corresponding Atari environment. The global trend shows monotonic improvement with model depth, driven by the prevalence of monotonic scaling regime tasks (12/26) (see Table 1) where over-parameterization is strictly beneficial. **(Right)** Training dynamics for Pong (the canonical regime). It reveals the canonical double descent trajectory: validation loss initially improves ($L = 2 \rightarrow 8$), deteriorates as it reaches the interpolation threshold ($L = 12 \rightarrow 48$), and recovers to achieve the lowest validation loss in the highly over-parameterized regime ($L = 96$). We present results averaged across eight independent runs and use a $\alpha$-trimmed mean filter with $\alpha = 0.2$ and a window size of 10. Error bars represent one standard error.

**Classical regime:** *Bigger models hurt.* In certain environments such as Amidar, the task complexity exceeds the effective capacity of even the largest model of $L = 96$ relative to the limited data to reach the interpolation threshold. While the training loss effectively reaches zero for models with $L \geq 4$, the validation loss exhibits a classical U-shaped curve. The intermediate models ($L = 8$ to $L = 12$) achieve the lowest validation loss, whereas further increasing depth to $L = 24, 48$, and $96$ leads to degradation in fidelity. The models remain in the classical overfitting regime; the interpolation threshold has not yet been surpassed, and thus, adding parameters is detrimental to generalization.

**Monotonic regime:** *Bigger is better.* Environments with moderate complexity, such as Assault, reside squarely in the modern over-parameterized regime. Here, we observe a monotonic improvement in validation performance as model depth increases. Unlike the high-complexity cases, even the smallest models in our sweep possess sufficient capacity to surpass the interpolation threshold. Consequently, further increasing the model size does not lead to overfitting; instead, the additional capacity serves to reduce variance and smooth the decision boundary, leading to strictly beneficial scaling behavior.

**Satuated regime:** *Small is already big.* Finally, low-complexity environments such as Freeway exhibit rapid saturation. Because the underlying dynamics are simplistic, even the shallower models (e.g., $L = 4$) effectively capture the data manifold. Consequently, the validation loss curves for larger models ($L = 8$ through $L = 96$) essentially overlap with no significant gain. In this regime, the task is solved by minimal capacity, rendering further scaling redundant though not harmful.

The right panels of Figure 3 depict the average loss curves aggregated across all 26 individually trained world models. For the complete set of empirical loss curves for every specific environment, refer to Figure 12 in the Appendix. The global average follows the trend of the monotonic regime, indicating that, in aggregate, the benefits of scaling outweigh the costs within our parameter sweep up to $L = 48$.

Notably, the left panels of Figure 3 reveal a distinct trend falling outside the previously defined categories:

**Canonical regime:** *Bigger hurts, then helps.* In environments exemplified by Pong, the task complexity is adequately positioned relative to our model sweep to manifest the full double descent phenomenon, spanning both classical and over-parameterized regimes. Initially, we observe standard classical behavior: increasing model depth from $L = 2$ to $L = 8$ reduces bias, leading to an improvement in validation loss. However, as we push beyond this local optimum into the critical regime ($L = 12$ to $L = 48$), the trend reverses. Here, the model capacity becomes just large enough to interpolate the training noise but is insufficient to smooth the decision boundary, causing variance to dominate and performance to deteriorate. Crucially, as

Table 1: Categorization of Atari environments by the generalization regime. We categorize 26 games into four regimes based on the alignment of the interpolation threshold relative to our model sweep ($L = 2 \ldots 96$) (see Figure 2). This reveals a spectrum of effective task complexity, ranging from HIGH (classical overfitting) to LOW (immediate saturation).

| Scaling regime | Count | Generalization Trend | Environments |
|---|---|---|---|
| CLASSICAL | 6 | *Bigger models hurt* | Alien, Amidar, Breakout, Gopher, Kangaroo, MsPacman |
| CANONICAL | 2 | *Bigger hurts, then helps* | Pong, Qbert |
| MONOTONIC | 12 | *Bigger is better* | Assault, Asterix, Boxing, ChopperCommand, CrazyClimber, DemonAttack, Hero, Jamesbond, Krull, PrivateEye, RoadRunner, UpNDown |
| SATURATED | 6 | *Small is already big* | BankHeist, Battlezone, Freeway, Frostbite, KungFuMaster, Seaquest |

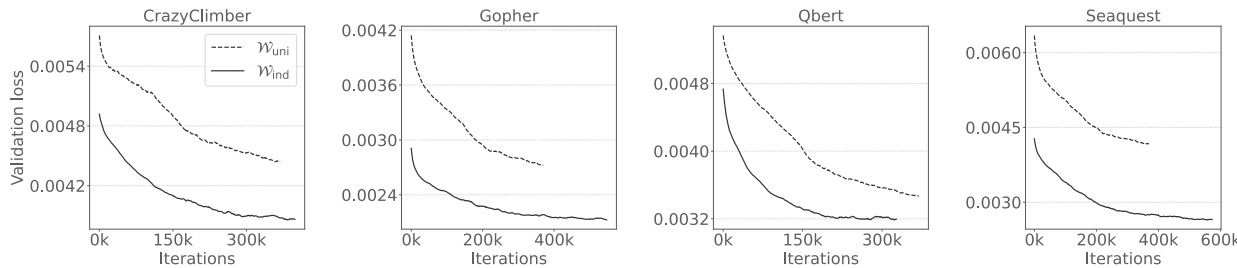

Figure 4: Validation loss comparison between the unified model $\mathcal{W}_{\text{uni}}$ (dashed) and individual models $\mathcal{W}_{\text{ind}}$ (solid) with fixed capacity ($L = 24$) across four representative environments. The specialist $\mathcal{W}_{\text{ind}}$ consistently outperforms the generalist $\mathcal{W}_{\text{uni}}$. This gap illustrates capacity dilution: whereas $\mathcal{W}_{\text{ind}}$ optimizes for a single manifold, $\mathcal{W}_{\text{uni}}$ must amortize its resources across 26 distinct distributions, thereby increasing error.

we further scale into the highly over-parameterized regime ($L = 96$), the validation risk declines once again, eventually surpassing the performance of the classical best ($L = 8$). This complete trajectory—where scaling is first beneficial, then detrimental, and finally restorative—justifies the designation of the *canonical double descent*, confirming that while intermediate scaling hurts, extreme scaling helps.

Table 1 provides a systematic categorization of all 26 Atari environments. We classify each game into one of the four generalization regimes based on the specific alignment of its interpolation threshold relative to our fixed model capacity sweep. The classification is based on the empirical loss curves of the separately trained world models on each environment. See Figure 12 for the environment-wise training results. Additionally, refer to Appendix D for our qualitative interpretation regarding the categorization of each gaming environment into distinct scaling regimes. Finally, we verify that the performance degradation observed in the classical regime is strictly attributable to traditional overfitting rather than training instability. While prior literature reports that scaling transformer depth can frequently lead to gradient divergence or attention entropy collapse (Li et al., 2020; Zhai et al., 2023), our deepest models ($L = 96$) consistently achieve near-zero training error across all environments (see Figure 2, bottom panels). This decoupling of training and validation trends confirms that the degradation is a result of high variance (generalization gap) rather than optimization failure. Lastly, to rule out the influence of stochastic noise, all reported curves in the main body represent the mean performance averaged over eight independent runs.

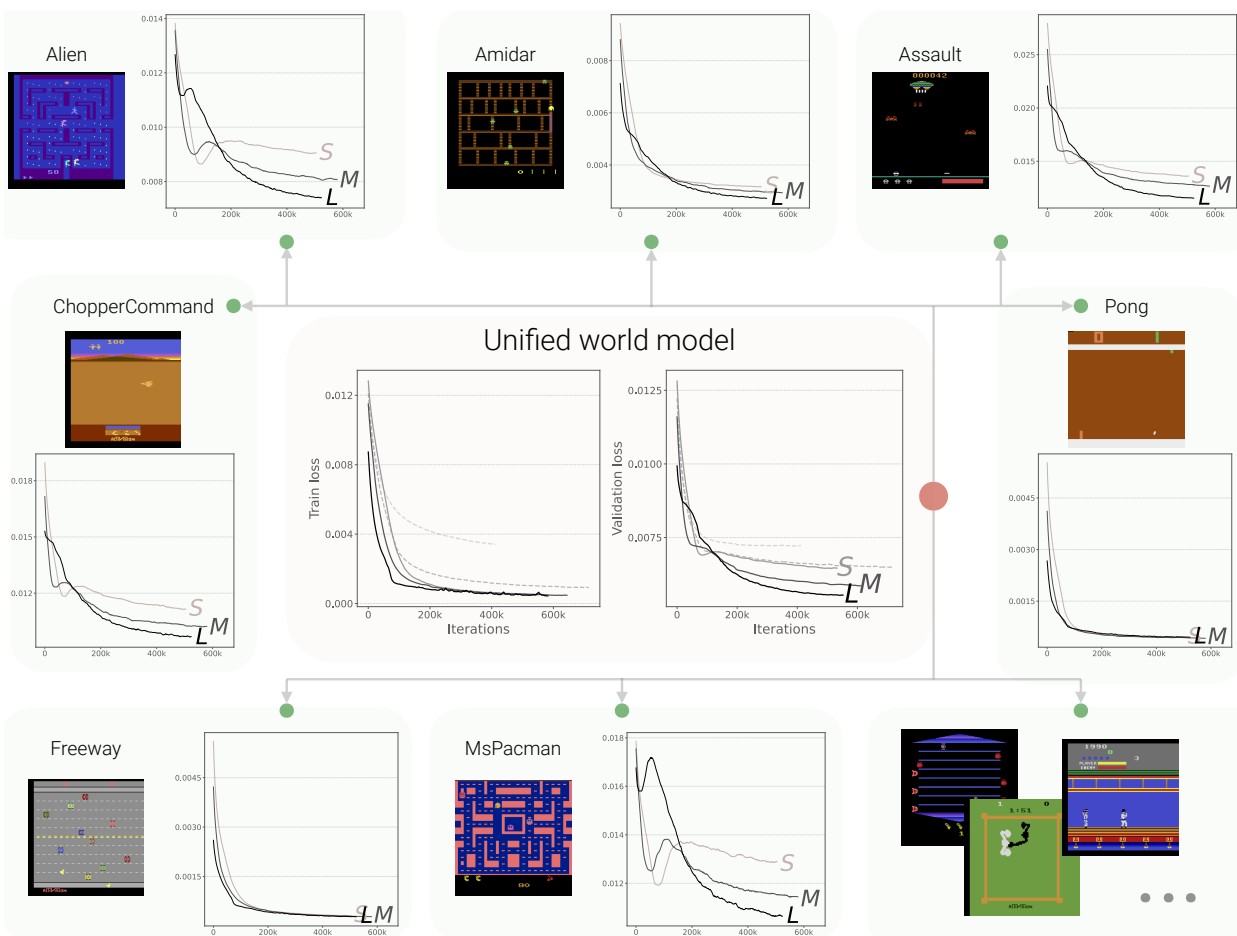

Figure 5: Learning dynamics of the unified world model $\mathcal{W}_{\text{uni}}$. We display aggregated and environment-wise learning curves for a single model instance trained simultaneously on all 26 Atari environments. **(Center)** Solid lines correspond to Small (S), Medium (M), and Large (L) configurations that possess sufficient capacity to reach the interpolation threshold, i.e., achieving near-zero training loss. Dashed lines represent underparameterized models, which converge to significantly higher loss values. **(Surrounding Panels)** Validation loss curves for 7 representative environments (selected from the full set of 26 due to space limits). We observe that within this unified training setup, performance monotonically improves or stabilizes with scale, confirming that *bigger models never hurt.* This stands in contrast to individual-environment training results, where increasing model capacity often leads to performance deterioration in environments with classical or canonical regimes, e.g., Alien, Amidar (Figure 2) or Pong (Figure 3).

## 4.3   Results on unified world models

We now evaluate the unified world model, $\mathcal{W}_{\text{uni}}$, a single transformer trained simultaneously on the aggregated dataset of all 26 Atari environments. Our objective is to determine how scaling laws evolve when transitioning from modeling isolated environments to a vast, heterogeneous joint distribution.

First, we quantify the performance trade-off between specialist and generalist models. Figure 4 contrasts the validation loss of the unified model against individual baselines across four representative environments with fixed capacity (latent dimensionality 512, 24 layers). We observe that $\mathcal{W}_{\text{ind}}$ consistently achieves lower validation loss than $\mathcal{W}_{\text{uni}}$. This phenomenon, known as *negative transfer* (Standley et al., 2020), is structurally expected: whereas the individual model dedicates its entire parameter budget to the idiosyncrasies of a single game, the unified model must amortize its capacity across 26 disjoint dynamical systems. Consequently, for a fixed parameter count, the generalist inevitably incurs a higher reconstruction error than the specialist.

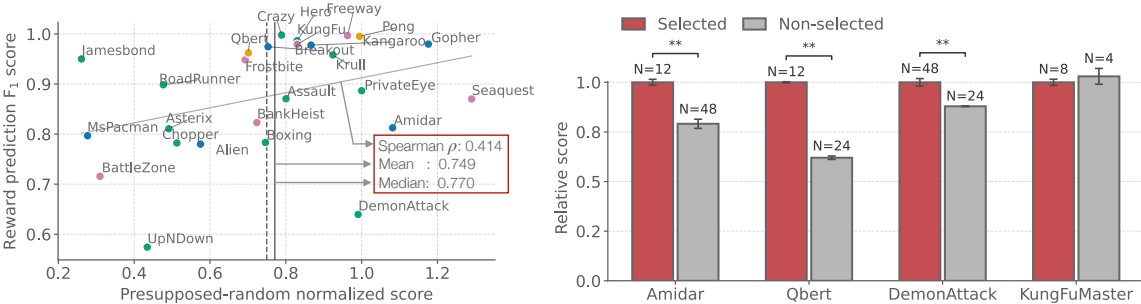

Figure 6: Downstream policy performance in learned environments. **(Left)** Normalized scores for 26 environments vs. reward predictor $F_1$ score. The mean (0.749) and median (0.770) scores confirm sufficient fidelity for agent training, while the correlation ($\rho = 0.414$) indicates performance is partly bounded by reward accuracy. **(Right)** Policy performance of optimally scaled (Selected) vs. baseline (Non-selected) models (mean $\pm$ SE over 16 independent trials). Other than for the saturated low-complexity case (KungFuMaster), matching model capacity to the inherent scaling regime yields significant gains ($p < 0.01$), confirming validation world model loss as a robust proxy for downstream policy-learning.

**Unified training: *Bigger models never hurt*** While the unified model incurs a performance penalty due to negative transfer, it offers a countervailing advantage through scaling stability. Figure 5 presents the learning dynamics of $\mathcal{W}_{\mathrm{uni}}$ across five model configurations, ranging from under-parameterized baselines ($L = 8$, $d \in \{512, 1024\}$; dashed lines) to over-parameterized networks ($L \in \{4, 8, 24\}$, $d = 2048$; solid lines, denoted as S, M, L, respectively), with the largest configuration reaching approximately 1.2 billion parameters. See Appendix G for the full results.

Once $\mathcal{W}_{\mathrm{uni}}$ enters the over-parameterized regime, we observe that bigger models consistently improve or sustain performance, evaluated on all environments. This stands in distinct contrast to the individual experiments ($\mathcal{W}_{\mathrm{ind}}$, subsection 4.2), where high- or intermediate-complexity environments (e.g., Alien, Pong) required precise architectural tuning to avoid entering the critical regime of degradation. In the unified setting, we observe that the joint distribution acts as a robust regularizer, effectively mitigating the inverted-U overfitting curve observed in isolated training. We posit that this reframes the standard multi-task learning narrative. While the literature often focuses on the detriment of *negative transfer*, our results highlight the distinct mechanism of the task diversity protecting individual environments from overfitting, supporting a "scale-first" strategy where model capacity can be increased with significantly reduced risk of degradation.

## 4.4 Results on policy learning in world models

Having established that appropriate scaling is crucial for enabling world models to faithfully mirror the real environment—either by tuning individual capacities to their scaling regimes or by training a unified model—we now address the logical subsequent question: does this improved fidelity translate to effective downstream policy learning? The central premise of our minimalist design is to emphasize the primacy of scale over structural sophistication. However, this hypothesis holds only if the resulting world models possess sufficient fidelity to support the training of competent agents. To rigorously test this, we evaluate PPO agents trained entirely within the *learned* dynamics of our best-performing individual world models. Crucially, the performance of these agents is bounded by that of the *presupposed* expert PPO policy used to collect the offline dataset. To ensure a strictly controlled comparison, we maintain the training configuration of the in-model agents—including network architecture, learning rate, and total optimization steps—identical to that of the expert presupposed agent, varying only the rollout horizon (32 vs. 128) while compensating with a quadrupled batch size to maintain parity in total environmental steps. To isolate the impact of transition dynamics from reward modeling, we deploy an external, independent reward predictor, also trained from the limited offline dataset, for each environment, held constant across all world model configurations (see Appendix E). This setup isolates the fidelity of the world model as the sole variable, allowing us to determine if properly scaled minimalist models can serve as effective surrogates for the real environment.

Figure 6 (left) summarizes the performance of policies trained within our best-performing individual world model configurations across all 26 environments. Refer to Appendix H for the learning curves inside the world models. We plot the normalized policy score (x-axis)—scaled such that 0 corresponds to a random agent and 1 to the presupposed expert—against the $F_1$ score of the external reward predictor (y-axis). The results demonstrate strong transfer fidelity: we achieve a median normalized score of 0.770 and a mean of 0.749, indicating that the minimalist world models successfully capture the critical dynamics required to recover approximately 75% of the expert's performance. Notably, in environments such as Amidar, Gopher, and Seaquest, the agents trained inside the world model exceed the performance of the presupposed policy used to generate the data. Furthermore, the scatterplot reveals a positive correlation (Spearman's $\rho = 0.414$) between reward prediction accuracy and policy performance. This suggests that in cases where performance lags, the limitation often stems from the external reward signal rather than the fidelity of the transition dynamics. Overall, these findings confirm that when scaled correctly to match the inherent scaling regimes of the environment, minimalist world models serve as effective surrogates for the real environment.

Finally, we examine whether the validation loss improvements achieved through specific scaling strategies translate into statistically significant differences in policy performance. Figure 6 (right) compares the normalized scores of policies trained in the optimally scaled world models (Selected) against those trained in standard baseline configurations (Non-selected) across four representative environments. For high-complexity environments (Amidar) and intermediate cases (Qbert), where smaller models ($L = 12$) outperformed larger baselines ($L = 48, 24$), we observe a corresponding significant advantage in downstream policy performance ($p < 0.001$). Conversely, for DemonAttack, where its inherent scaling regime supports the *Bigger is Better* trend, the larger model ($L = 48$) significantly outperforms the medium baseline ($L = 24$). The only exception is the low-complexity environment KungFuMaster, where performance saturates early; here, both the small ($L = 8$) and very small ($L = 4$) models achieve similar degree of expert recovery. These results, averaged over 16 independent trials with error bars representing the standard error, confirm that the scaling regimes identified via validation loss are predictive of downstream task success. This underscores that correct model scaling is not merely a metric-optimization exercise but a prerequisite for learning functional policies.

## 5   Discussion and Conclusion

In this work, we isolated the impact of scale in transformer world models, demonstrating that performance is not merely a function of architectural priors but is strictly governed by the interplay between model capacity and task complexity. Our analyses identified distinct scaling regimes: individual models frequently encounter performance degradation in complex environments due to capacity saturation, exhibiting an inverted-U curve. While large-scale foundation models rely on massive data abundance to safely bypass these capacity bottlenecks and guarantee monotonic scaling, we demonstrate that this predictable scaling can be recovered even under strict data constraints. Specifically, we found that unified training across diverse environments provides a powerful regularizing effect. This task diversity mimics the stabilizing benefits typically achieved through immense data volume, effectively shifting the scaling dynamics and ensuring monotonic improvements across all tasks regardless of their individual complexity. Notably, preliminary results (Appendix I) suggest that this stabilization persists even under a strictly fixed training budget, highlighting a promising avenue for deeper study into the interplay between generalist diversity and scaling stability. We further confirmed that this improved fidelity directly enables high-performing policies learned in imagination.

Despite these monotonic improvements, scaling alone does not yet definitively overcome the persistent negative transfer inherent in multi-task learning, as individual specialist models still act as empirical lower bounds. Future research must rigorously investigate this interplay between massive scale and multi-task interference, alongside extending the analysis to environments necessitating long-horizon reasoning. Moreover, while this work focused on offline datasets, a truly generalist agent must operate in a conventional iterative setting: progressing from random actions to world model formulation, policy optimization, and subsequent data collection in a continuous loop. This shift introduces the challenge of non-stationary data distributions, where maintaining the model's ability to adapt becomes paramount; effectively mitigating the loss of plasticity (Dohare et al., 2024) will be essential for sustained learning. Finally, achieving a single, unified controller will likely require scaling the policy networks in tandem with the world model to master diverse dynamics across diverging environments.

**Acknowledgments**

This work was supported by Institute of Information & communications Technology Planning & Evaluation(IITP) grant funded by the Korea government(MSIT) (No.RS-2020-II201336, Artificial Intelligence Graduate School Program(UNIST)) and partly supported by the Institute of Information & Communications Technology Planning & Evaluation (IITP) grant funded by the Korea government(MSIT) (No. RS-2025-25441313, Professional AI Talent Development Program for Multimodal AI Agents, Contribution: 50%).

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

# A    Details on collecting the presupposed environmental steps using PPO

We adopt a backcasting-based research design; we start by presupposing well-performing reinforcement learning policies, trained directly from the Atari environments with hundreds of millions of environmental steps, and use them to collect non-random, quality 400K environmental steps, or, equivalently, 100K (observation, action, reward and termination) triplets. Then, using the collected trajectories, we formulate world models that closely mimic the dynamics of the actual gaming environment. When the world models are well-configured and well-trained, the policies learned inside the world models' imaginary MDPs can be expected to perform comparably to the benchmark performance, which is the performance achieved by the presupposed policy [1]. The collected trajectories using the presupposed policies should include episodes with human expert-level returns that undergo multiple stages within games. At the same time, those should also comprise some degree of stochasticity to account for diverse patterns of environmental dynamics.

## A.1    Model and training configurations

To this end, we manage pre-trained policy networks for different games to collect episodic trajectories using the proximal policy optimization (PPO) (Schulman et al., 2017) algorithm. The policy training results of the PPO algorithm on the Atari environments are disclosed in the original paper. However, we choose to train the presupposed PPO policies from scratch for the following reasons: First, to the best of our knowledge, the full algorithm checkpoints for all 26 environments under the same algorithmic configurations are not disclosed. For example, the original paper does not include the training results of the environment Hero. Second, we are interested in finding out to what extent the policies learned solely inside the world model can recover the returns earned from the presupposed policies that are learned directly from the environments with hundreds of millions of environmental steps. Therefore, it is indispensable to use the same parameteric configurations for both the world model policies as well as the real-world, presupposed policies, to suppress potential unappreciated biases that can arise from different model structures and configurations. Figure 7 shows the comparisons of the mean episodic returns of the original PPO paper and the newly trained presupposed policies in this paper. While the error bars for the original paper's results are not disclosed, the authors report that the reported numbers are means over a hundred episodes per environment.

We use the generalized advantage estimation (GAE) to approximate the advantage function. The value tensor has shape $(B, T+1)$ to accommodate the bootstrap value at the final time step, enabling computation of temporal-difference (TD) errors between consecutive states. The training configuration is listed in Table 2, and the model structure of the presupposed PPO is detailed in Table 3. The observation images are normalized with a mean of 0.5 and a variance of 0.5. We use early stopping with respect to the average of the last 100 running returns, with the patience parameter set to 300. The maximum iteration is set to 50,000, and therefore, the maximum number of environmental steps is approximately 100 million ($50,000 \times 128$ (horizon) $\times 16$ (batch size)).

## A.2    Environmental setup

For each environmental step, we collect a quadruplet of observation, action, reward, and termination signal. To train the presupposed PPO, the observation frames are collected as RGB $64 \times 64$ images. Unless otherwise noted, we follow the settings of the original PPO paper (Schulman et al., 2017); we use a constant frame-skip parameter of 4; we regard an episode as terminated when a single life is lost. [2]; we set the max noop (no operation) start to 30 to add minimal stochasticity to episodic rollouts. Figure 8 illustrates the learning curves of the presupposed PPO algorithms with respect to the 26 gaming environments.

---

[1]We assume that the transformer world model cannot imagine beyond its training experience

[2]In some games, such as Breakout or Frostbite, a single episode consists of multiple lifes and the episode termination signal is enabled only when the whole lifes are consumed.

### A.3 Collection of the trajectories

We collect 100K environmental steps per games using the presupposed PPO policy algorithm. At this stage, the goal is to make the world models formulated from the collected trajectories mimic *all* actual game environmental dynamics as close as possible. If we use random actions only to collect the full 100K steps, the world model will immediately collapse when the agent inside the world model advances to stages that are not reachable from random actions, due to the lack of generalizability or the extrapolation capability of the transformer world models. On the other hand, if we solely rely on the presupposed PPO, the imaginary MDP will also collapse when agents deviate from the learned policy. To account for the both ends, we collect environmental trajectories with gradually decreasing stochasticity. Specifically, for each collection step $0 \leq i < 100,000$, the agent takes random actions with probability $p = 1 - (\log_{10} (1 + i) \div 5)$. We assume $p = 1$ if $p > 0.99$. Otherwise, the agent takes an action sampled from the PPO algorithm. The collection results are listed in Figure 9.

| Configuration | Value |
|---|---|
| Input dimensions | 64x64x3x4 |
| Input method | Frame stacking (4) |
| Discount ($\gamma$) | 0.99 |
| GAE parameter ($\lambda$) | 0.95 |
| Surrogate clip coef. | 0.2 |
| Entropy coef. | 0.005 |
| Value coef. | 0.5 |
| Value clip coef. | 0.1 |
| Clip gradient norm | 0.5 |
| Horizon (T) | 128 |
| Optimizer | Adam |
| Learning rate | $2.5 \times 10^{-4} \times \alpha$ |
| Target KL | 0.01 |
| KL coef. | 1.5 |
| # Epoch | 4 |
| VectorEnv size (B) | 16 |
| Minibatch iter (MI) | 8 |
| Minibatch size | 256 ($T \times B/MI$) |
| Max. # iterations (I) | 50000 |
| Max. # Env. steps | $\simeq$ 100M ($T \times B \times I$) |

Table 2: Training configuration for presupposed PPO. $\alpha$ is linearly annealed from 1 to 0 over the course of learning.

| Structure | | Value |
|---|---|---|
| Activation | | ELU |
| Backbone convolutional layers | Layer 1 | Kernel size (8x8), Stride 4 |
| | Layer 2 | Kernel size (4x4), Stride 2 |
| | Layer 3 | Kernel size (3x3), stride 1 |
| Actor head | | 1 hidden layer (512x512) |
| Critic head | | 1 hidden layer (512x512) |

Table 3: Model structure specifications for presupposed PPO.

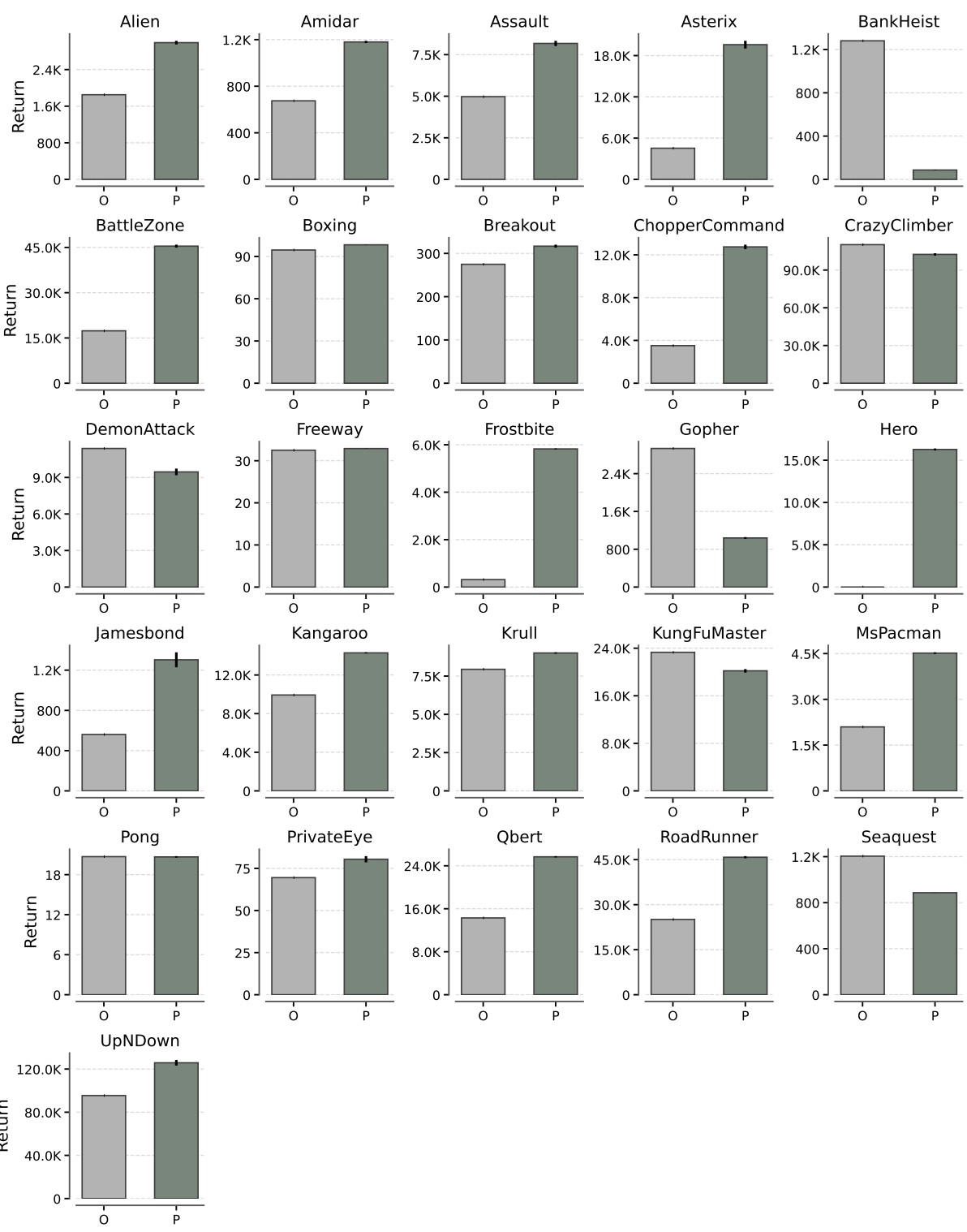

Figure 7: Comparisons of the training results of the original PPO paper (O; left; gray) and the newly trained presupposed policies in this paper (P; right; green). For the presupposed results, the error bars represent one standard error over at least 120 episodes. For the game Hero, the original results are not available and therefore left blank. In this figure, episodic life loss is set to false and a single episode has multiple lives for the fair comparison with the original PPO results.

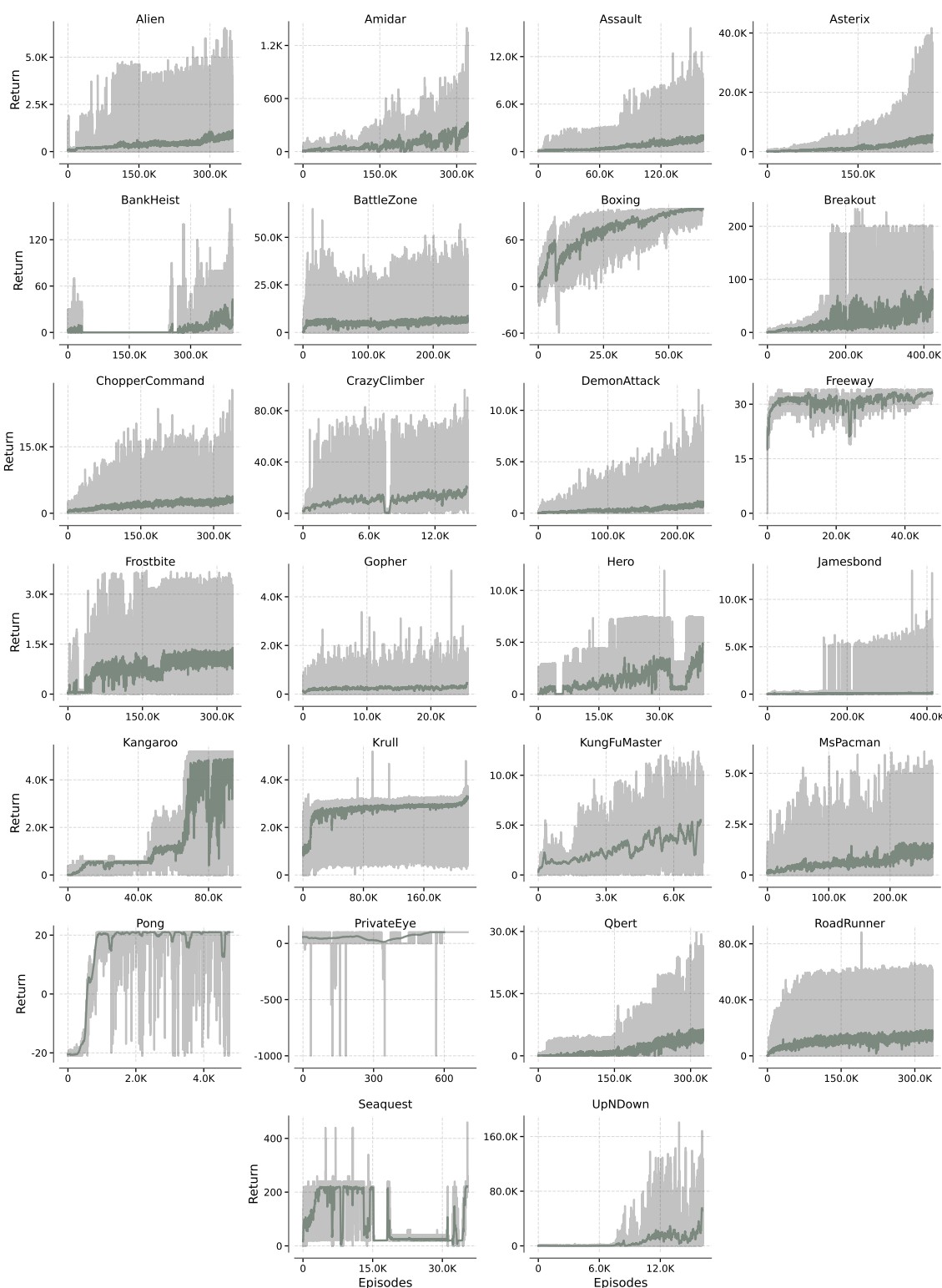

Figure 8: Learning curves of the presupposed PPO algorithms on 26 Atari 100K benchmark environments. We report the running returns of the games with an episodic-life-loss setting. The gray lines show the complete learning curves whereas the green lines represent the results after the $\alpha$-trimmed mean filtering with $\alpha$ set to 0.1 and the window size set to 100.

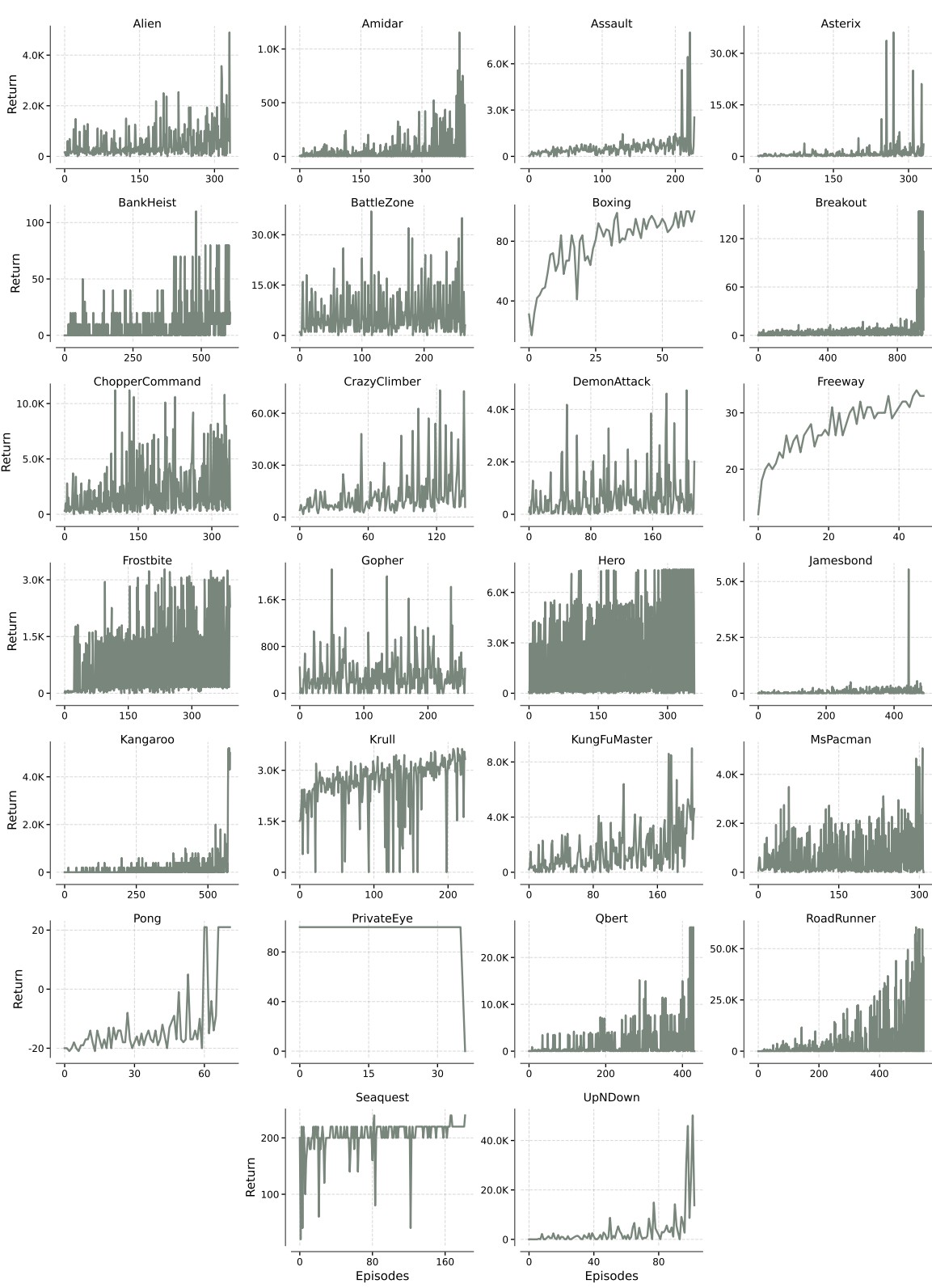

Figure 9: Collection of the presupposed trajectories with 100K environmental steps. We gradually decrease the randomness of the actions.

# B  Variational autoencoder configurations and training results

We use a convolutional neural network (CNN) (LeCun et al., 1998)-based variational autoencoder (VAE) (Kingma & Welling, 2014; Rezende et al., 2014) for the world model's vision component. Our implementation is based on a publicly available CNN-VAE repository (Ditria & Drummond, 2023) with an additional perceptual loss using the pretrained VGG16 model (Simonyan & Zisserman, 2014). For the benefit of simplicity, we train a single VAE model on all 26 Atari gaming environments. Unless otherwise noted, we use RGB input images of shape: $64 \times 64 \times 3$, which results in the latent embedding dimensionality of 512 under the current VAE configuration. We use early stopping with respect to the validation loss, with the patience parameter set to 100. Table 4 lists the hyperparameter settings used throughout the paper. Figure 10 illustrates the validation mean squared errors (MSEs) during training procedures. Figure 11 display the observed and reconstructed (encoded and then decoded) image frames by the VAE models trained on 26 Atari gaming environments.

## B.1  Preserving the visual details in the vision component

Preserving and maintaining the visual details within the world model rollouts is crucial for the returns earned within the world model to be seamlessly translated into the real environments. In this subsection, we introduce three different architectures used as the visual-sensory component to process observation frames in their respective world models used in regard to the Atari 100k benchmark.

To this end, numerous model architectures are proposed. The Dreamer models by Hafner et al. (2021; 2025a), the Transformer World model (TWM) by Robine et al. (2023), and another transformer-based world model, STORM, by Zhang et al. (2023) use categorical encoder-decoder structures with the straight-through gradient estimation method suggested by Bengio et al. (2013). Note that this approach is strictly distinctive from the categorical VAE method (Maddison et al., 2017; Jang et al., 2017) that leverages the Gumbel-Max trick in that it does not conduct the posterior inference over the latent representation variables. Additionally, all the aforementioned models with the categorical encoder-decoder structure for the visual-sensory component are operationalized under the end-to-end joint learning scheme, i.e., the gradient from the visual-sensory component flows through the memory component, e.g., transformer backbone.

IRIS (Micheli et al., 2023) and DART (Agarwal et al., 2024) resort to vector-quantized variational autoencoder (VQ-VAE) (Van Den Oord et al., 2017) to represent a single image into a fixed-sized discrete sequence of tokens. Subsequently, large-scale world models with implications for being deployed in real-world physical environments include such vector quantization approaches (Hu et al., 2023; 2024). This line of approach is directly analogous to the language modeling using transformers in that an image corresponds to a sentence comprising multiple discrete tokens, i.e., words. Aside from the loss of information during the discretization process, the vector quantization approach offers a direct tradeoff between the reconstruction quality and the generation of imaginary image observations, i.e., the number of function evaluations (NFE), by adjusting the number of tokens assigned per image.

Lastly, Alonso et al. (2024) propose the diffusion model (Ho et al., 2020) as their world model backbones. Here, the generation process of a future image is directly conditioned on the previous image pixels without the latent variable bottleneck.

Conducting a comprehensive analysis on how the choice of the model architecture for the visual-sensory component affects the overall capability of the world model is a prominent future direction. Furthermore, to the best of our knowledge, the benefit of the joint learning of both the vision as well as the memory (dynamics) components is yet to be strictly evaluated. Hafner et al. (2021) compares the performance of their discrete encoder-decoder structure against the Gaussian latent variable counterpart, but within their framework of joint-learning and recurrent state-space model (RSSM).

In Figure 11, we observe that the visual details necessary to learn meaningful policies within the world models are captured with high fidelity. For example, all the eggs represented as small dots in the Alien (top left) environment and in similar games such as BankHeist and MsPacman are faithfully captured in the reconstructed images. Also, all the broken blocks as well as their exact positions are captured in the Breakout environment (top right). Lastly, in the game Asterix (fourth from the top left), the objects that

Table 4: VAE hyperparameter settings.

| Hyperparameter | Value |
|---|---|
| Latent channels | 32 |
| Channel multiplier | 64 |
| Block sizes | (1, 2, 4, 8) |
| Batch size | 26 * 8 |
| Optimizer | Adam |
| KL scaler value | 1.0 |
| Perceptual loss scaler value | 1.0 |
| Max. # iterations | 2M |
| Clip gradient norm | 1.0 |
| Activation function | ELU |
| Evaluation interval | 1,000 |
| Evaluation # iterations | 100 |
| Learning rate | $1.0 \times 10^{-4}$ |

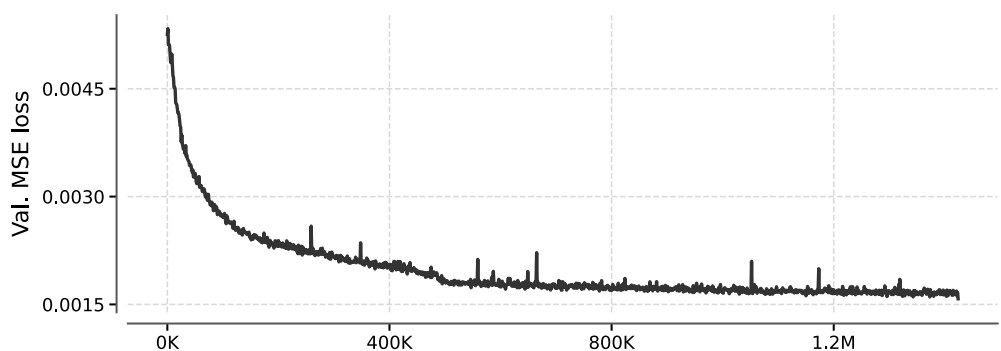

Figure 10: MSE loss on validation sets for the VAE models. The size of the input image is $64 \times 64 \times 3$, and the latent dimensionality is 512.

need to be avoided and the ones that need to be taken are challenging to distinguish, with the same color and similar shapes. We observe that such subtle differences are also correctly reflected in the reconstructed observation images. The difficulty of capturing the visual details in the Atari games, specifically in the aforementioned environments, is highlighted in the work of Alonso et al. (2024).

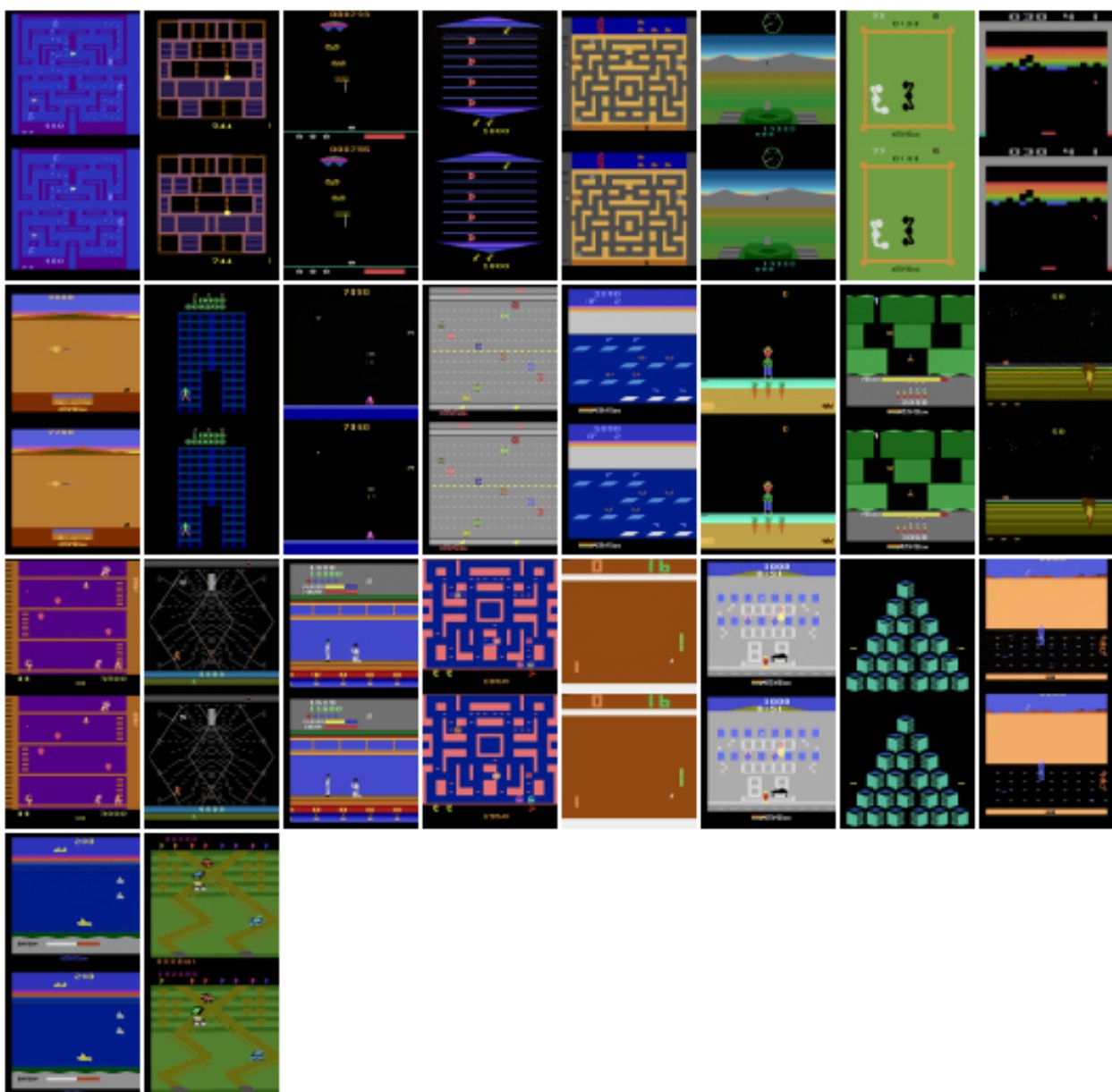

Figure 11: VAE training results with the original observations at the bottom and the reconstructed (encoded-decoded) images at the top. Input images have sizes $64 \times 64 \times 3$ and the embedding dimensionality is 512.

Table 5: Configuration details for the world model backbone. Here, each transformer world model is separately trained for a single Atari environment.

| Parameter | Value |
|---|---|
| Architecture | GPU-like Transformer (decoder-only) |
| Normalization | LayerNorm (per layer) |
| Input Representation | VAE Latents (continuous) |
| **Vocabularies** | |
| Action Space | Discrete, Environment-dependent |
| Reward Space | 6 tokens |
| | (Sign $\{-1, 0, 1\} \times$ Termination $\{0, 1\}$) |
| **Dimensions** | |
| Embedding Dim. | 512 |
| Attention Heads | 16 |
| Context Length | 432 tokens ($144 \times 3$ triplet steps) |
| Dropout | 0.0 |

## C   Transformer world model configurations and training results

Table 6: Configuration details for the world model training.

| Hyperparameter | Value |
|---|---|
| **Optimization** | |
| Optimizer | AdamW ($\beta_1 = 0.9, \beta_2 = 0.95$) |
| Learning Rate | $1 \times 10^{-4}$ (Constant, No Warmup) |
| Batch Size | 16 |
| Weight Decay | 0.1 |
| Gradient Clipping | 1.0 |
| **Objective** | |
| Loss Function | $\mathcal{L}_{\text{obs}} + \alpha \cdot \mathcal{L}_{\text{reward}}$ |
| Observation Loss | MSE (on VAE latents) |
| Reward Loss | Cross Entropy |
| Reward Loss Weight ($\alpha$) | $1 \times 10^{-5}$ |
| **Training Loop** | |
| Max Iterations | $1,000,000$ |
| Evaluation Interval | 1,000 iterations |
| Early Stop Patience | 100 evaluations (100k steps) |

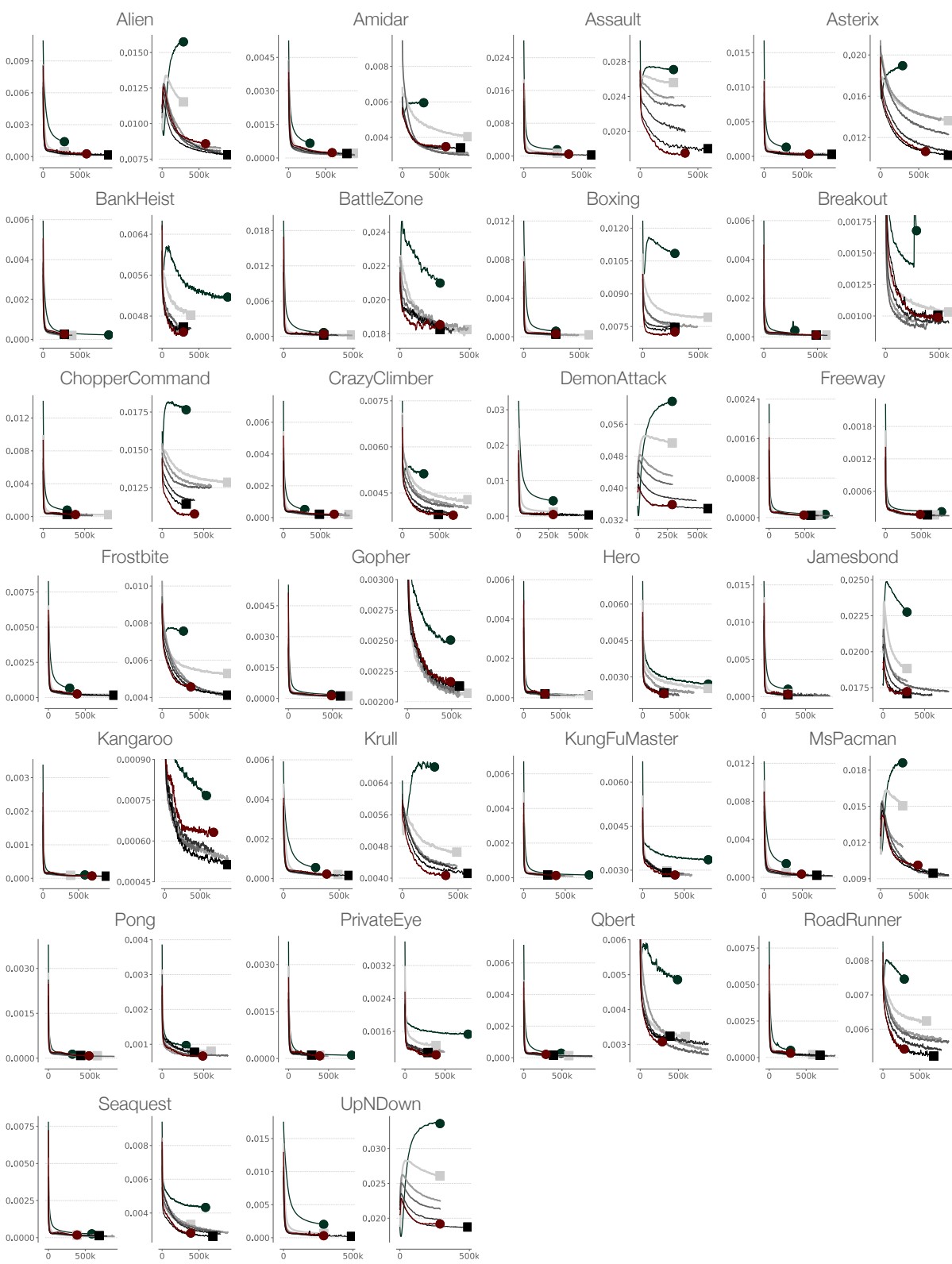

Figure 12: Empirical loss curves of the transformer world models, each of which was separately trained on 26 Atari games. For each game, the left and right panels depict the training and validation losses, respectively. Here, we use $\alpha$-trimmed mean filtering with $\alpha = 0.2$ and a window size of 10.

# D    Qualitative interpretation of distinctive scaling regimes

We offer an interpretive framework for why specific environments fall into their respective parameterization regimes, as summarized in Table 1. In the context of world modeling, "complexity" (or intrinsic dimensionality) refers not to the difficulty of playing the game, but to the difficulty of *predicting the next frame* given the limited interaction history. This distinction explains why visually simplistic games can exhibit high intrinsic dimensionality while visually rich games may be effectively low-dimensional.

**Classical regime**    Games in this regime (Bigger models hurt) typically involve dense maze logic, independent enemy agents, or chaotic collision physics. Alien, Amidar, and MsPacman require the world model to track not only the player's position but also the global state of the maze and the independent, often multimodal logic of multiple enemies. Predicting the next pixel frame requires implicitly learning the "rules" of the maze boundaries; a model that fails to capture these constraints will hallucinate agents passing through walls. Breakout presents a unique case of dynamical chaos. While visually sparse, the transition dynamics rely on precise elastic collisions. A minute error in the world model's estimation of the ball's angle results in a diverging trajectory that compounds rapidly. Further, the speed of the ball movement changes drastically according to the location of the brick, i.e., hitting the upper, previously covered bricks results in higher instant rewards and faster ball movement after the reflection. In this data-restricted regime (100k frames), the task complexity exceeds the capacity of the model to learn the generalized physics, causing larger models to overfit to specific trajectories in the replay buffer rather than interpolating the underlying dynamics.

**Canonical regime**    These environments sit at the critical threshold of our model sweep ($L = 2 \dots 96$), exhibiting the Canonical double descent. Pong is conceptually similar to Breakout but involves simpler physics (fewer collision objects, no destructible bricks). Consequently, the complexity is low enough that our largest models ($L = 96$) can successfully bridge the interpolation gap, unlike in Breakout. The intermediate models ($L = 12 \dots 48$) capture the noise in the bounce dynamics but lack the capacity to smooth the decision boundary, leading to the characteristic peak in validation loss before the final descent.

**Monotonic regime**    This category comprises the majority of the benchmark (12 games) and follows the modern scaling law where *Bigger is Better*. Games like Assault (scrolling shooter) or Hero involve dynamics that are complex but structured. Unlike the chaotic collisions of Breakout or the rigid maze constraints of Amidar, the dynamics here are often continuous and locally consistent (e.g., projectiles moving in arcs, scrolling backgrounds). The model capacity here directly translates to better tracking of these objects and smoother interpolation of movement, allowing the validation loss to decrease monotonically with depth.

**Saturated regime**    Environments in this regime (*Small is already big*) are characterized by linear, independent, or deterministic dynamics. Freeway is the archetype of this category: the agent moves vertically, and cars move horizontally at constant speeds. The prediction of the next frame effectively requires memorizing a linear translation function ($x_{t+1} = x_t + \Delta$). Similarly, KungFuMaster features a linearly scrolling corridor with predictable enemy spawns. Because the underlying manifold of valid next-states is so low-dimensional, even the smaller models ($L = 4$) possess sufficient capacity to fully capture the dynamics, rendering further scaling redundant.

# E   Details on external reward and termination predictors

To decouple reward and termination estimation from world model scaling, we train standalone, external predictors for each environment. This design choice ensures that any observed differences in policy performance stem solely from the fidelity of the learned dynamics, rather than variations in reward modeling across different configurations or scaling regimes. By holding the reward signal constant, we rigorously isolate the impact of the world model's imagination quality. These predictors are trained using the fixed offline dataset.

**Data Preprocessing.** Following standard practice in Atari 100k, we employ reward clipping, mapping all positive rewards to $+1$, negative rewards to $-1$, and zero rewards to 0. Consequently, the reward predictor is formulated as a 3-class classifier, while the termination predictor is a binary classifier. However, other than Pong and Boxing, the reward signal only consists of 0 and 1 without -1.

**Architecture.** Both the reward and termination predictors share a similar architecture, conditioning on a history of observations, rewards, termination signals, and an action. Specifically, we encode the inputs into three distinct feature streams: (1) The VAE latent embeddings of the past four observations are concatenated and processed via an MLP; (2) The previous action token is converted to a one-hot vector and embedded via a second MLP; (3) The sequence of the past four reward signals is processed by a third MLP. These three latent representations are concatenated and fed into a final MLP classification head. For reward prediction, the head outputs logits for the clipped reward classes $\{-1, 0, 1\}$. For termination prediction, it outputs logits for the binary termination signal.

**Training & Selection.** Both models are optimized using the standard Cross-Entropy loss. Given the extreme sparsity of rewards and termination events in Atari games, we address class imbalance through class weighting and label smoothing (sweeping $\alpha \in [0.01, 0.1, 0.2]$). For the reward predictor, model selection is strictly based on maximizing *precision* for the positive reward class. This criterion is critical for stability in imagination: a false positive (predicting a reward where there is none) creates a "delusion" that the agent can exploit, leading to catastrophic policy divergence in the real environment. Conversely, a false negative is a more benign failure mode in this context. A similar precision-maximization strategy is applied to the termination predictor to prevent premature episode endings in imagination.

# F    Comparison of the existing transformer world models in Atari

Table 7 details the diverse architectural configurations employed by existing transformer world models in Atari. Implementations vary significantly in scale, ranging from shallow, wide networks (STORM: 2 layers, 512 embedding dim) to deeper, narrower models (TWM, IRIS: 10 layers, 256 embedding dim), with parameter counts spanning from 6M to 200M. Tokenization strategies typically alternate between categorical encoders (TWM, STORM) and VQ-VAEs (IRIS), while sequence modeling approaches differ in their treatment of actions and rewards—either integrating them as autoregressive tokens (TWM, Decision Transformer) or handling them via separate mechanisms (STORM). These variations in depth, width, context length, and state representation underscore the lack of a standardized scaling regime in current literature.

Table 7: Configuration comparisons of transformer world models in Atari

| | TWM | IRIS | STORM | Decision transformer |
|---|---|---|---|---|
| Layers | 10 | 10 | 2 | 4, 6, 10 |
| Embedding dimension | 256 | 256 | 512 | 512, 768, 1280 |
| Attention heads | 4 | 4 | 8 | 8, 12, 20 |
| Latent representations | Categorical-encoder | VQ-VAE | Categorical encoder | N/A (patching) |
| Agent state | Training: Latent Inference: Obs. (frame stacking) | Reconstructed Obs. (CNN-LSTM) | Latent + hidden | N/A |
| Actions as tokens | TRUE | TRUE | FALSE (Action mixer) | TRUE |
| Reward as tokens | TRUE | FALSE | FALSE | TRUE |
| Sequence length | 16 | 20 | 64 | 4 |
| Imagination horizon | 15 | 20 | 8 | N/A |
| Total # parameters | 19M | 8M | 6M | 10M, 40M, 200M |

# G  Additional results on unified world model

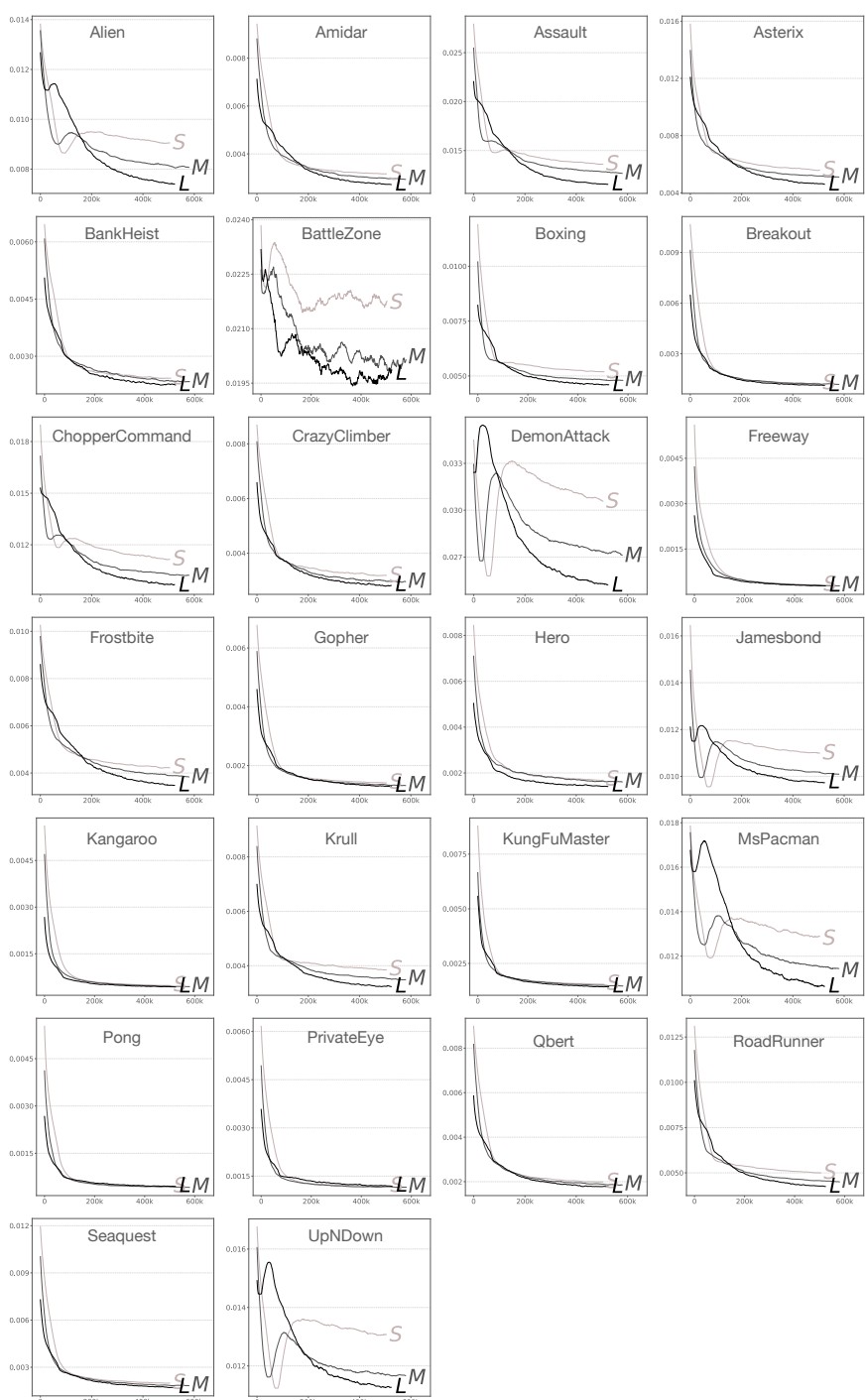

Figure 13: Environment-wise results for the unified world model. In stark contrast to the individual setting, we observe a consistent scaling behavior across all environments: the validation loss decreases monotonically from the Small (S) to the Medium (M) and Large (L) configurations, or otherwise saturates. Crucially, we observe no rank reversals or inverted-U curves, empirically confirming that the unified training regime stabilizes scaling dynamics.

## H    Learning curves for the policy learning in world models

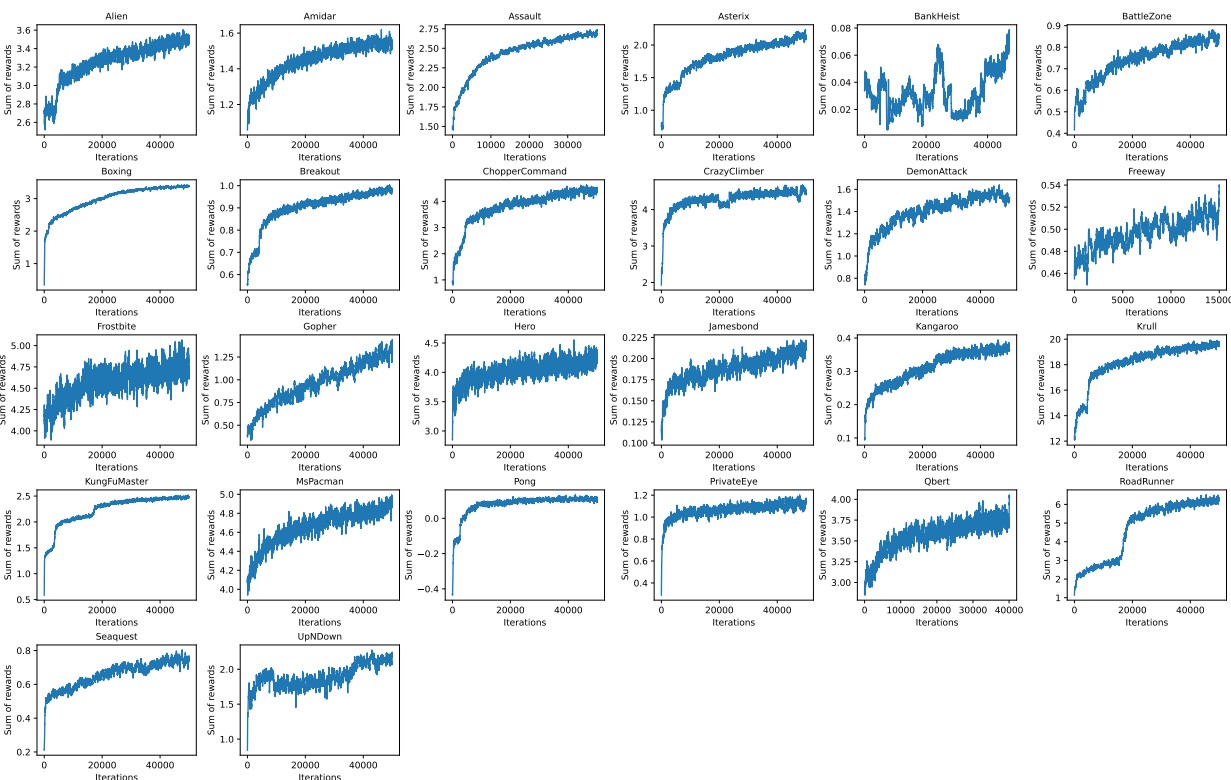

Figure 14: Learning curves for the policy learning in world models.

# I   Unified training with 100k budget

To investigate whether task diversity contributes to the stabilization of scaling dynamics independently of the total data volume, we conducted a preliminary ablation study. We trained models on a strictly fixed budget of 100k frames, comparing a single-environment baseline (100k frames of Amidar) against two distinct mixed-environment datasets: Amidar combined with Assault (50k frames each) and Amidar combined with Gopher (50k frames each).

As shown in Figure 15, allocating half of the data budget to a different environment noticeably alters Amidar's scaling curve. In the 100k Amidar-only baseline, validation loss spikes early, transitioning from $L = 12$ to $L = 24$. However, when mixed with Assault—an environment belonging to the monotonic scaling regime—the capacity-induced overfitting spike is substantially delayed. The validation loss for Amidar continues to decrease up to $L = 48$, only deteriorating at $L = 96$ (Figure 15A, B). The inclusion of Assault's dynamics effectively shifts Amidar's scaling trajectory, making it significantly less characteristic of the classical regime and more monotonic.

In contrast, mixing Amidar with Gopher—another environment residing in the classical regime—produces a much milder delay in the overfitting spike (Figure 15C, D). While these observations are preliminary and restricted to two environmental pairs, they provide a valuable glimpse into how integrating diverse dynamics can alter relative model capacity and mitigate overfitting, even under strictly constrained data budgets. This highlights the interplay between task complexity and scaling regimes as a promising direction for future rigorous investigation.

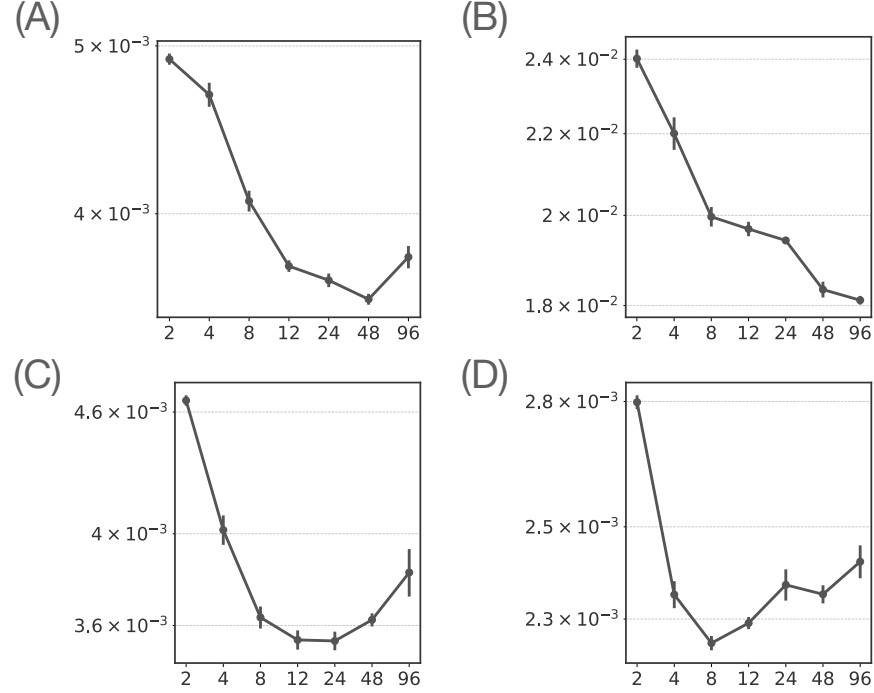

Figure 15: Scaling behavior under a strictly fixed total data budget of 100k frames. The x-axis denotes model capacity (number of layers, ranging from 2 to 96), and the y-axis represents the minimum validation loss achieved via early stopping. (A, B) Performance on a 50k/50k mixture of Amidar (classical regime) and Assault (monotonic regime). Integrating the monotonic environment significantly delays Amidar's overfitting spike, resulting in a more monotonic scaling curve. (C, D) Performance on a 50k/50k mixture of Amidar and Gopher (both classical regime), which yields a notably weaker regularizing effect on Amidar's scaling dynamics. Error bars inticate one standard error over four independent runs.

