# OpenReview forum: "Probing the Impact of Scale on Data-Efficient, Generalist Transformer World Models for Atari"
_TMLR — Accepted by TMLR_

### Review · Reviewer_1xfr · 2026-02-08

**Summary Of Contributions:**

The paper proposes the scaling behavior of world models trained on RL environments depends on the intrinsic dimensionality of the task. For simple tasks, bigger models help, while for complex tasks, scale actually hurts performance. This is explained by the double descent phenomenon. However, models trained on all tasks (unified models) always improve with scale. This is explained by the task unification yielding a regularization effect, thereby mitigating the double descent error spike. This improvement with scale translates from unified world models to policies.

**Audience:**

Yes

**Audience Explanation:**

Absolutely- the question of when scaling positively/negatively impacts performance is critical, particularly as a function of task complexity. This is especially important in the context of world models, where this direction remains unexplored. The paper also touches on whether unified world models are better than specialized world models, which is also a very relevant and timely question in the community.

**Broader Impact Concerns:**

No broader impact concerns.

**Claims And Evidence:**

No

**Claims Explanation:**

The paper explains the transition in scaling behavior for low vs. high complexity tasks in terms of the double descent curve: the claim is that for complex tasks, models are typically in the underparameterized regime, and larger models approach the interpolation threshold, thereby performing worse. This is an interesting hypothesis, but this needs to be more evidence behind this claim. A telltale sign of the double descent interpolation threshold is that the error spike corresponds to the point where the *training error* first gets very small. However, looking at Figures 2 and 3, this doesn't seem to obviously be the case. Perhaps this would be easier to see if the y-axis for the training loss plots were on a log scale.

Another important question is at what point in training this double descent behavior emerges. Typically, by early stopping, the double descent error spike can be somewhat avoided. It's unclear whether this is the case looking at Figure 2: perhaps zooming into iterations near zero or using a log scale on the x-axis would help.

I may have missed this, but there doesn't seem to be a quantitivate evaluation of the intrinsic dimensionality of each task.This is critical; otherwise, the classification in Table 1 appears heuristic. The explaination in Appendix D is too qualitative to be very useful unfortunately. At the very least, there should be some quantitative measure of the complexity of each task (if not intrinsic dimensionality).

It's unclear to what extent some of the results are statistically significant. For instance, Figures 2 and 3 (and Figure 12 in the Appendix) don't include error bars over multiple trials.

**Requested Changes:**

**Critical**
- Quantitatively evaluate intrinsic dimension of each task
- Use evidence from the training loss to support the double descent explaination
- Error bars over multiple trials for main results (or some other uncertainty quantification)

**Would strengthen**
- Explain how training time/early stopping affects the emergence of double descent behavior
- In Figure 2, instead of using distinct colors for different model choices, it would be better to use a color gradient indicating small to big
- Text in Figure 4, 6 is too small

---

> ### Author Response · Authors · 2026-02-14
> **Response to reviewer 1xfr**
>
> We thank reviewer `1xfr` for taking the time and effort to review our paper and providing insightful comments. Your comments and suggestions are addressed in the revised manuscript, and we summarize the changes we made below:
>
> > **P1. Training loss and double descent:**: *"[...] for complex tasks, models are typically in the underparameterized regime, and larger models approach the interpolation threshold, thereby performing worse. [...] A telltale sign of the double descent interpolation threshold is that the error spike corresponds to the point where the training error first gets very small. However, looking at Figures 2 and 3, this doesn't seem to obviously be the case."*
>
> - We thank the reviewer for this keen observation. We agree that a rigorous examination of the training loss is crucial to accurately situate different Atari environments within specific regions of the deep double descent landscape.
> - We respectfully highlight that in our setting of the generative world models, the "interpolation threshold" differs from the "zero training **error**" seen in deterministic classification tasks. Within our objective of predicting the continuous VAE latents in stochastic environments, the training loss is bounded by the aleatoric uncertainty, and achieving the zero training **loss** (MSE) is infeasible.
> - Following your recommendation, we have updated Figures 2 and 3 in the main manuscript to plot the training and validation losses on log-scales. This new visualization reveals:
>     - The training loss drops by 1--2 orders of magnitude across all scaling regimes, whereas the validation loss drops at a significantly slower rate, or deteriorates in complex tasks.
>     - The training loss saturates at or before $L=8$ for all environments. Beyond this point, increasing model capacity yields no significant reductions in training loss.
> - With this context, our results map precisely to the standard deep double descent landscape. For the complex environment of Amidar (Figure 2, left-most panels):
>     - At $L < 2, 4$ lies the classical underparametrized regime, with both training and validation loss decreasing.
>     - At $L = 8$, the training loss hits the noise floor (saturates; interpolation threshold).
>     - For larger model capacities, the train loss remains saturated, indicating the model has fully captured the data up to the aleatoric limit, but the validation loss begins to rise.
> - We believe the updated log-scale figures clarify that our observations are indeed consistent with the double descent literature.
> - **Action taken:** We have replaced Figures 2 and 3 in the main text with the log-scale versions to make this behavior explicit for future readers.
>
> > **P2. Early stopping and the emergence of double descent**: *"[...] Typically, by early stopping, the double descent error spike can be somewhat avoided. It's unclear whether this is the case looking at Figure 2."*
>
> - We clarify that an early stopping criterion was indeed employed for all experiments to ensure fair comparison (patience: 100 evaluations; see Appendix C). For the complex task (Amidar),  the minimum validation loss achieved by each model during training, not the loss at a fixed final epoch, worsens for the model capacities beyond the interpolation threshold.
> - The fact that the performance degradation persists even with optimal early stopping is consistent with the previous findings. As demonstrated in the original deep double descent paper (Nakkiran et al., 2019; Figure 19) and further analyzed by Pezeshki et al. (2022), model-wise double descent can occur even when each individual model is early-stopped at its peak performance.
> - We observe epoch-wise double descent in the learning curves (e.g., $L=4$ in Amidar). This confirms that the critical regime manifests both structurally (across model sizes) and temporally (during training), reinforcing that our observed scaling behaviors are driven by the fundamental capacity-data mismatch rather than optimization artifacts.
> - **Action taken:** Following your suggestion, we have updated the learning curves in Figures 2 and 3 to use a log-scale x-axis. This visualization allows for a closer inspection of the early training dynamics.
>
> *Pezeshki, Mohammad, et al. "Multi-scale feature learning dynamics: Insights for double descent." International Conference on Machine Learning. 2022.

---

> ### Author Response · Authors · 2026-02-14
> **Response to reviewer 1xfr (2)**
>
> > **Q3. Quantitative evaluation of the classification (Table 1)**: *"[...] I may have missed this, but there doesn't seem to be a quantitative evaluation of the intrinsic dimensionality of each task. This is critical; otherwise, the classification in Table 1 appears heuristic."*
>
> - We appreciate the reviewer pointing out this confusion. We clarify that the classification in Table 1 was not heuristic (i.e., it was not based on a qualitative assessment of game mechanics as described in Appendix D). Instead, the classification was strictly empirical, derived post-hoc from game-wise quantitative learning curves in Figure 12 of Appendix C. For example, games were assigned to the "classical regime" (formerly "high intrinsic dim") if their validation loss diverged at higher capacities, and games were assigned to the "monotonic regime" (formerly "medium intrinsic dim") if their validation loss improved monotonically.
> - We acknowledge that using the term "intrinsic dimensionality" implied a causal, a priori measurement of complexity, which made the classification appear circular. To rectify this, we have renamed the categories in Table 1 and throughout the text to "scaling regimes" (classical, canonical, monotonic, saturated). This terminology accurately reflects that the grouping is a result of the observed scaling behaviors, not an input hypothesis.
> - **Action taken**: We have revised the text to explicitly state that Appendix D serves only as a qualitative interpretation of why certain games might fall into these empirically identified regimes, rather than serving as the classification criteria itself. Throughout the paper, we have replaced the term "intrinsic dimensionality" with "scaling regimes".
>
> > **Other points**:
>
> - We added the error bars that signify one standard error for figures 2 and 3.
> - We changed the color schemes to a grayscale color gradient.
> - We enlarged the font size for Figures 4 and 6.

---

### Review · Reviewer_26BZ · 2026-02-23

**Summary Of Contributions:**

The premise of the paper is straightforward: if we do our best to simplify the model architecture, what is the impact of world model scale on a benchmark like Atari 100k?

The authors find that, even though models trained on individual games fall into distinct categories of scaling behaviors (sometimes failing to scale), models trained on multiple games do not fall to their lowest common denominator, and instead enter a predictable scaling regime consistent with literature on overparameterized models. They also find that these simple multi-game world models are suitable world models to train agents on (with PPO).

**Additional Comments:**

I think this is good science, just because it looks backwards and asks, "ok what did we actually learn?". Yet, this paper feels on the edge of something greater, and while I believe that this paper is totally fine for TMLR, I do think a stronger paper could easily be a "big 3" conference paper.

**Audience:**

Yes

**Audience Explanation:**

I think this is a relevant paper in that it's a useful data point for work on scaling & world models.

**Broader Impact Concerns:**

No broader impact statement is provided.

**Claims And Evidence:**

Yes

**Claims Explanation:**

The central claims are well backed by extensive multi-seed runs, I have no doubt that they are solid results.

I do think thought that the _breadth_ of evidence in the paper is lacking. The central claim is proven, with more games and more data we enter the conventional ("canonical") deep learning scaling regime. But why?

Some (in my opinion) simple experiments are missing:
- is the cause simply more data? What happens if a set of 100k / 26 frames _per-game_ are chosen (i.e., training data size is kept fixed)? Conversely, what happens if a model is trained on 2.6M frames for a non-scaling ("classical") games? (since you have the base agent, you should be able to generate this)
- is the cause a regularization of dynamics? What if you just paired two environments, one scaling ("monotonic") and one non-scaling ("classical"), would you still see improvement? Which behavior "wins"? (again ideally keeping total #data fixed) What about two or more non-scaling environments?
- maybe it's there but I'm not reading the text/figures correctly, but what happens beyond a fixed parameter count in Figure 4? Does scale eventually beat negative transfer or do individual models act as lower bounds?

**Requested Changes:**

Just one thing, the lines in Figure 5 aren't super legible, please consider re-rendering

---

> ### Author Response · Authors · 2026-03-10
> **Response to reviewer 26BZ**
>
> We sincerely thank Reviewer `26BZ` for the highly encouraging review. We are thrilled that you found our core claims solid and our approach to be good science. Your insightful questions regarding the underlying mechanisms of the unified model directly motivated our additional experiments during the discussion period.
>
> >**P1. Disentangling Data Volume from Task Diversity**: *"[...] is the cause simply more data? What happens if a set of 100k / 26 frames per-game are chosen [...] is the cause a regularization of dynamics? What if you just paired two environments, one scaling ("monotonic") and one non-scaling ("classical") [...] What about two or more non-scaling environments?"*
>
> - We found your proposed ablations highly compelling. To isolate the effect of task diversity from sheer data volume, we constructed two new mixed offline datasets with a strictly fixed budget of 100k frames (mirroring the single-game budget):
>     - Classical + Monotonic: 50k frames of Amidar + 50k frames of Assault.
>     - Classical + Classical: 50k frames of Amidar + 50k frames of Gopher.
>
> - We hesitate to make definitive claims because doing so would require more extensive, rigorous, and carefully conducted studies. However, it does seem that the regularization of dynamics plays a certain role. Even without increasing the total data volume, environmental diversity delayed capacity-induced overfitting for Amidar. In the 100k Amidar-only baseline, validation loss spiked early, transitioning from $L=12 \to 24$. In the mixed Amidar + Assault setting, the loss continued decreasing up to $L=48$, with the spike delayed until $L=96$. The Amidar + Gopher mixture also delayed the overfitting spike, though the regularizing effect was milder (for Gopher as well).
> - This signals that leveraging diverse dynamics through unified training might act as a task regularizer aside from adding more data.
>
> - **Actions Taken**: We have added the fixed-budget mixture experiments to the Appendix. We reference them in the main text to offer a glimpse into the unified model's scaling behavior on stabilizing the scaling regimes across different tasks in the absence of data abundance and to motivate further investigation.
>
> > **P2. Scaling Limits and Negative Transfer:** *"[...] what happens beyond a fixed parameter count in Figure 4? Does scale eventually beat negative transfer or do individual models act as lower bounds?"*
>
> - We briefly investigated this scaling behavior beyond the fixed parameter count. While each individual environment exhibited its own distinctive pattern, looking at the overall average across all 26 environments revealed no significant change in the performance gap. We revised the discussion to note this observation and highlight future research on the interplay between stabilization through model scale and negative transfer.

---

### Review · Reviewer_7Lmr · 2026-02-24

**Summary Of Contributions:**

This work seeks to understand the relationship between performance and scale in the context of world modeling. The authors generate a fixed dataset from a PPO trained model across 26 Atari tasks, then use that dataset to train world models with differing numbers of layers with a similar training algorithm. There are models trained on individual tasks, as well as all the tasks simultaneously.

The authors find that the validation performance versus scale (and more generally, the training curves) for each of the tasks can be interpreted as lying in different regions of the "canonical" double descent curve --- where model performance is initially decreasing in capacity, then starts increasing, and then decreases again. By combining the training and validation loss information at the end of training, the authors argue that the chosen model families interpolate different regions of the curve for different tasks.

Finally, the authors claim that the combined model exhibits "positive regularization" where the performance improves monotonically over the model family.

**Audience:**

Yes

**Audience Explanation:**

I believe that these results would be interesting to readers working in optimization, who do not have as much experience in RL/world models.

However, I do believe that some points are either oversold or not properly contextualized. This is reflected in my requested changes section, but I believe these comments can be addressed with some minor rewrites.

**Claims And Evidence:**

Yes

**Claims Explanation:**

I think the overall idea of the study is sound. I have not seen other studies of this type in this area, though I am more familiar with the optimization literature on double descent, and less on the reinforcement learning/world modeling literature. I have some suggestions for better improving the presentation of the arguments that different parts of the double descent curves are covered by different tasks, but looking at the plots provided I do agree with the arguments.

There are two major issues with the paper:

1. In much of modern machine learning at scale, particularly the "foundation models" referred to in the introduction/motivation, it appears that most modeling setups are in a regime where performance increases monotonically with model size, when reasonable training procedures are used. This is some combination of large, complex datasets/tasks as well as a better understanding of how to choose optimization parameters and take advantage of implicit and explicit regularization. This is not sufficiently acknowledged in the motivation or the discussion, and does to some extend limit the impact of this work.

2. The concept of "positive regularization" is needlessly misleading; using the authors' own motivation, their results on the unified world model can be simply explained by the fact that they decreased relative model complexity by increasing data/task complexity. I think the paper should be very explicit about this point, and not muddle the literature with unnecessary terminology.

**Requested Changes:**

Figure 2 should include plots of final validation loss vs number of layers, so that readers do not just have to rely on the cartoons to see the various stages of double descent.

Figure 3 is currently confusing, since only pong has the associated regime cartoon. In general, I think the cartoon figures of the different regimes should be separated a bit more from individual plots, and the authors should rely on direct plotting of the validation losses to show their points with actual data.

Another feature which should be compared in main figure plots is the gap between the train loss and validation loss. In the "classical"/underparameterized regime you should be able to see a small generalization gap; this increases towards the critical regime, and decreases to an asymptotic-but-non-zero constant in the overparameterized/data limited regime. There should be evidence of train loss nearing its minimum possible value only in the overparmeterized regime (which I think it does in the figures; is the minimal training loss $0$ in this case?)

The nomenclature of "positive regularization" seems unnecessary. The results on the unified model can be more easily explained by the fact that the data/task complexity has increased while the model complexity (or more accurately, model complexity family) remains unchanged. Given that some subset of the individual tasks are in the classical regime, it is not at all surprising that the combined tasks are in the classical regime as well. This extra terminology makes a simple phenomenon seem more deep than it actually is. In my opinion this is a crucial change.

There should be overall more acknowledgment that in many complex, modern settings (e.g. "foundation models"), with good training setups performance is generically monotonic in model size over most practical ranges of parameters.

---

> ### Author Response · Authors · 2026-03-10
> **Response to reviewer 7Lmr**
>
> We thank reviewer `7Lmr` for the time and care taken to review our paper and for the helpful feedback. We have addressed the comments in the revised manuscript and summarize the changes below:
>
> > **P1. Contextualizing with modern foundational models:** *"[...] There should be overall more acknowledgment that in many complex, modern settings (e.g. "foundation models"), with good training setups performance is generically monotonic in model size over most practical ranges of parameters.", "[...] This is not sufficiently acknowledged in the motivation or the discussion, and does to some extend limit the impact of this work."*
>
> - We agree that acknowledging the monotonic scaling of modern foundation models does indeed strengthen the motivation of our work. By explicitly stating that foundation models bypass capacity-induced performance degradation through massive data and compute, we can better highlight the penetrating message of our paper, which focuses on analyzing the scaling dynamics in strictly data-constrained environments where the reliance on massive data is removed.
> - **Actions taken**:
>     - Introduction: We have devoted the first paragraph of the introduction to explicitly acknowledging this paradigm. We now detail how foundation models achieve monotonic scaling through immense data abundance, advanced optimization procedures, and careful regularization (adding corresponding citations: [A-H]). We then directly contrast this with the strictly data-efficient setting of the Atari 100k benchmark to correctly frame our study's impact.
>     - Discussion: We have updated the discussion section to reflect this insight at the conclusion of the paper. We now explicitly frame our multi-task findings through this lens, noting that the task diversity in our unified model effectively mimics the stabilizing benefits typically achieved through data volume in foundation models, allowing us to recover predictable, monotonic scaling under data limits.
>
> > **P2. The nomenclature of "positive regularization"**: *"[...] The concept of 'positive regularization' is needlessly misleading; using the authors' own motivation, their results on the unified world model can be simply explained by the fact that they decreased relative model complexity by increasing data/task complexity. [...] This extra terminology makes a simple phenomenon seem more deep than it actually is. In my opinion this is a crucial change."*
>
> - We agree with your critique. While our intention was to highlight the stabilizing effect of multi-task learning, we recognize that introducing a new term obscures the fundamental mechanics at play. As you noted, this behavior is cleanly explained by standard statistical learning: relative model capacity decreases as overall task complexity increases.
> - We were curious to explore whether task diversity itself contributes to this stabilization independently of data volume. To that end, we ran a small, preliminary experiment (motivated by the suggestion from the reviewer `26BZ `).
> - We trained a model on a strictly fixed budget of 100k frames (50k from Amidar [classical regime] and 50k from Assault [monotonic regime]). Even without increasing the total data volume, this mixture delayed capacity-induced overfitting for Amidar. In the 100k Amidar-only setting, validation loss spiked when transitioning from $L=12$ to $L=24$. In the mixed setting, the loss continued decreasing up to $L=48$, only spiking at $L=96$.
> - A similar, though milder, delay occurred when mixing Amidar with Gopher, another classical environment.
> - Given the limited scope of this experiment (only two environment mixtures and targeting a certain environment only), we are hesitant to make definitive claims. However, it suggests the possibility that environmental diversity might offer a regularizing effect that goes beyond simply increasing the dataset size. We have included these preliminary results in the Appendix to motivate future investigation.
> - **Action taken**: We have globally removed the phrase "positive regularization" from the manuscript, following the suggested mechanistic framing. Also, we have addressed the potential future direction of investigating how unified training without total data volume can result in stabilizing the environments that were previously located in the classical regime, with the additional experimental result added in the appendix.

---

> ### Author Response · Authors · 2026-03-10
> **Response to reviewer 7Lmr (2)**
>
> **P3. Analyzing the Train-Validation Generalization Gap**: *"[...] Another feature which should be compared in main figure plots is the gap between the train loss and validation loss... is the minimal training loss 0 in this case?"*
>
> - We agree that explicitly visualizing the generalization gap clarifies the scaling dynamics. We have updated Figures 2 and 3 to plot both training and validation losses on a log scale. As hypothesized, these plots now clearly show the expanding and contracting gap across the classical, critical, and overparameterized regimes.
>
> - Regarding the minimal training loss: it saturates as the model hits the interpolation threshold, but it does not reach absolute zero. Because our objective involves predicting continuous VAE latents in stochastic Atari environments, the training loss is bounded by the environment's irreducible aleatoric uncertainty (the noise floor), causing it to asymptote to a non-zero constant.
>
> - **Actions Taken:** We have updated Figures 2 and 3 to include log-scale training loss curves, visualizing the generalization gap across regimes. Also, we have added clarification that the asymptotic training loss represents the environmental noise floor rather than zero.
>
> **P4. Replacing Cartoons with Data Plots:** *"[...] Figure 2 should include plots of final validation loss vs number of layers... I think the cartoon figures of the different regimes should be separated a bit more from individual plots..."*
>
> - Thank you for this suggestion. We have removed all regime cartoons from Figures 2 and 3. To explicitly demonstrate the scaling regimes with actual data, we have replaced them with plots showing the validation loss versus model capacity (number of layers). To ensure a fair comparison across scales, we plot the minimum validation loss achieved via the early stopping criterion, rather than the strictly final loss at the end of training.
>
> [A] Kaplan et al., "Scaling laws for neural language models", arXiv, 2020
>
> [B] Hoffmann et al., "Training compute-optimal large language models", In NeurIPS 2022
>
> [C] Loshchilov and Hutter, "Decoupled Weight Decay Regularization", In ICLR 2019
>
> [D] Yang et al., "Tuning large neural networks via zero-shot hyperparameter transfer", In NeurIPS 2021
>
> [E] You et al., "Large Batch Optimization for Deep Learning: Training BERT in 76 minutes", In ICLR 2020
>
> [F] Neyshabur et al., "In search of the real inductive bias: On the role of implicit regularization in deep learning", arXiv, 2014
>
> [G] Soudry et al., "The implicit bias of gradient descent on separable data", JMLR, 2018
>
> [H] Lin et al., "Scaling laws in linear regression: Compute, parameters, and data", In NeurIPS 2024

---

### Decision · Action_Editor_yr2Z · 2026-04-05

**Recommendation:** Accept as is

**Additional Comments:**

Please make sure that all updates mentioned in the rebuttal are in the paper.

**Audience:**

Yes

**Audience Explanation:**

The paper is looking at world models under limited data regime, trying to understand the dynamics of scale. Particularly they look at how within a multitask regime, there is an interplay between how for different task the model is either in the classic regime or double descent regime. I think such questions could be of interest to those either interested in optimization or world models.

**Claims And Evidence:**

Yes

**Claims Explanation:**

Yes, all reviewers of the submission seem satisfied with the evidence provided. While more experiments can only help, and reviewers complaint particularly regarding the original submission, the extra data points brought up during rebuttal give sufficient support to the claims.